# LAYER-AWARE INFLUENCE FOR ONLINE DATA VALUATION ESTIMATION

## ABSTRACT

Data-centric learning emphasizes curating high-quality training samples to boost performance rather than designing new architectures. A central problem is to estimate the influence of training sample efficiently. Prior studies largely focus on static influence measured on a converged model, overlooking how data valuation dynamically changes during optimization. This omission neglects the dynamic nature of sample influence during optimization, especially in deep models. To address the computational burden of frequent influence estimation, we develop a layer-aware online estimator that requires only loss-to-output gradients. This design avoids parameter-level and full-network gradients while preserving ranking fidelity. Extensive experiments across LLM pretraining, fine-tuning and image classification demonstrate that our method improves accuracy with substantially lower time and memory cost in both text and image datasets, making dynamic data curation both efficient and scalable in practice.

## 1 INTRODUCTION

Data-centric learning surges as an emerging topic in machine learning community, which focuses on curating high-quality training samples for performance boosting, rather than designing novel algorithms (Koh and Liang, 2017; Sedova et al., 2023; Ash et al., 2019). Sample influence estimation, also known as data attribution or data valuation, is the fundamental research question in data-centric learning, which assesses the sample importance associated with a certain model. Based on that, detrimental samples can be identified, which will be removed from the training set for another round training with the expected performance gain.

Sample influence estimation can generally be categorized into two categories (Hammoudeh and Lowd, 2022). 1) Retraining-based methods include the classical leave-one-out influence approach (Cook and Weisberg, 1982) that retrains models and observes performance changes after removing one training sample. While useful as an ideal baseline, this is *computationally untenable* on deep models. Other representative methods such as model-agnostic Shapley value approaches (Ghorbani and Zou, 2019; Jia et al., 2019; Kwon and Zou, 2022) also suffer similar problems. Computationally efficient approaches, such as KNN-Shap (Jia et al., 2018), are limited to using KNN classifiers and their recursive calculations. 2) Gradient-based methods can be used to approximately estimate influence without expensive overheads of retraining. Influence functions (Koh and Liang, 2017; Schioppa et al., 2022; Yang et al., 2024) are representative tools in this category, which employ the first-order Taylor expansion to estimate the performance change with sample gradients. Although model retraining is avoided, the inverse of the Hessian matrix in influence functions brings great challenges for large deep models. Several pioneering studies focus on approximating the inverse of the Hessian matrix in an efficient way (Koh and Liang, 2017; Grosse et al., 2023; Kwon et al., 2023).

While the above pioneering studies shed light on model-associated sample influence for identifying beneficial or detrimental samples in terms of data quality, they overlook another crucial factor—online data valuation dynamically change during optimization, particularly in deep models. This introduces a critical limitation: samples identified as detrimental based on the initial model may no longer be detrimental for the newly trained model after their removal. Such oversight not only necessitates two rounds of training but also results in suboptimal performance and fails to harness the full potential of online data valuation-aware training.

**Contributions**. To address the aforementioned limitations, we introduce a layer-aware online data valuation framework that estimates per-sample influence during training and integrates naturally with SGD-style updates. Our estimator aligns scoring with optimization and can be instantiated as policies such as online curation, reweighting, or priority sampling—without a filter–then–retrain cycle. Our key contributions are summurized as follows:

- *Research Question:* We formalize online data valuation that jointly accounts for data quality during model optimization, addressing the scoring–optimization mismatch and order-dependent interactions that static methods ignore.

- *Technical Innovation:* We design a Hessian-free, layer-aware online influence estimator that backpropagates only to the model outputs, avoiding full-network parameter gradients. Our lightweight calibration mitigates cross-layer scale bias while preserving valuation fidelity, enabling single-run training with minimal overhead and seamless integration with SGD.

- *Experimental Validation:* Extensive experiments are conducted across diverse scenarios, including LLM pre-training, LLM fine-tuning, and both image and text classification. The results demonstrate that our proposed method is highly effective and also computationally efficient.

## 2 RELATED WORK

In this section, we introduce the data valuation with a focus on influence function, and other online data curation topics including curriculum learning and data scheduling.

**Influence function**. Influence functions, originally developed in robust statistics (Hampel, 1974; Cook and Weisberg, 1982; Martin and Yohai, 1986), quantify the sensitivity of model parameters to perturbations in training data. Introduced to the machine learning community by Koh and Liang (2017), they have since been widely applied to tasks such as identifying detrimental sample, detecting outliers, and spotting mislabeled samples. Using a first-order Taylor expansion, influence functions estimate sample influence based on the model's performance on a validation set, leveraging the inverse of the Hessian matrix and sample gradients—eliminating the need for model retraining.

However, computing the inverse of the Hessian matrix is computationally prohibitive, particularly for large deep models. Several pioneering studies address this challenge by approximating the inverse of Hessian matrix more efficiently. LiSSA (Koh and Liang, 2017) employs Hessian-vector products for approximation, while EKFAC (Grosse et al., 2023) incorporates efficient eigen decomposition techniques. DataInf (Kwon et al., 2023) further simplifies influence calculations for large models by substituting the Hessian inverse with a rank-1 closed-form expression.

Beyond the static estimation of sample influence based on a final model (sometimes referred to as a checkpoint), recent approaches focus on dynamic influence estimation. GEX (Kim et al., 2024) leverages geometric ensembles of multiple checkpoints to approximate influence functions, mitigating bilinear constraints and addressing non-linear losses. Similarly, TDA (Bae et al., 2024) introduces a checkpoint-based segmentation strategy that combines implicit differentiation and unrolling, leveraging EKFAC (Grosse et al., 2023) for efficiency. TRAK (Park et al., 2023) employs randomly-projected kernel to reduce the dimensional of the Hessian matrix for tractable computing. Recently, a surge of interest in Hessian-free influence functions has emerged, which replaces the Hessian inverse with a simple identity matrix (Charpiat et al., 2019; Pruthi et al., 2020; Yang et al., 2024; Killamsetty et al., 2021). These approaches offer computational simplicity and scalability, making them particularly attractive for deep learning models with competitive performance.

While the above studies have made significant advancements in data-centric learning, they largely overlook the dynamic variance of sample influence throughout the model training process. Although some recent works attempt to address dynamic influence estimation, their approach—simply summing up sample influences at different checkpoints—fails to capture the evolving nature of sample influence effectively. This limitation arises because model parameters at early checkpoints often differ substantially from those at later stages, rendering static aggregation methods inadequate. Furthermore, these studies neglect the role of online data valuation change during the optimization, which is particularly critical for deep models. By treating data valuation as static, existing methods miss opportunities to dynamically adapt training strategies based on evolving sample importance, ultimately compromising both efficiency and effectiveness in influence-aware learning.

Most influence-function methods define the downstream objective on a held-out validation set, which is natural in supervised learning. However, large-scale pre-training and unsupervised regimes often lack such a labeled validation split. To handle these settings, recent work studies self-influence, which replaces the validation set with the training set itself and computes per-example influence scores based only on training loss and gradients (Bejan et al., 2023). They show that self-influence provides a stable and useful signal for data curation in the absence of a separate validation set, improving downstream machine translation, question answering, and text classification.

Beyond the above Hessian-based and Hessian-free formulations, several recent works revisit influence estimation from an optimization-aware and non-convex perspective. SGD-Influence (Hara et al., 2019) explicitly tracks the stochastic gradient descent trajectory and estimates the effect of removing a training example by reusing intermediate iterates, thereby relaxing the local convexity assumption and avoiding a single stationary point. MoSo (Tan et al., 2023) further models data pruning in deep networks by approximating a moving-one-sample-out update along the optimization path, while Z0-Inf (Kokhlikyan et al., 2025) develops a zeroth-order estimator that bypasses explicit gradients and Hessian information. These methods provide important progress toward making influence analysis more faithful in non-convex settings, but they are still instantiated as static data valuation procedures: they aggregate information over a completed training trajectory and then apply a filter–then–retrain pipeline. In contrast, our work focuses on online influence estimation within each optimization step, and designs a layer-aware, Hessian-free estimator that aligns scoring with SGD-style updates so that data valuation can directly guide training without a second pass.

**Curriculum learning**. Curriculum learning (Bengio et al., 2009; Wang et al., 2021; Soviany et al., 2022) is a training strategy that incorporates data sequence into model optimization by presenting data in an easy-to-hard order, mimicking the structured learning process observed in human education. As a general training paradigm, curriculum learning has been widely applied across diverse domains, including supervised learning tasks in computer vision (Guo et al., 2018; Jiang et al., 2014) and natural language processing (Platanios et al., 2019; Tay et al., 2019), as well as reinforcement learning (Florensa et al., 2017; Narvekar et al., 2017; Ren et al., 2018). It has also been extended to specialized applications such as graph learning (Gong et al., 2019; Qu et al., 2018) and neural architecture search (Guo et al., 2020). Despite its conceptual appeal, curriculum learning does not always yield performance improvements in machine learning tasks (Kumar et al., 2010; Zaremba and Sutskever, 2014; Hacohen and Weinshall, 2019). We hypothesize that two primary reasons may explain this limitation: The predefined easy-to-hard curricula are often human-designed and may not be optimal for model training. Easy samples may contribute minimally to building robust models, whereas hard samples may be noisy or even detrimental to performance, raising questions about their actual utility. To address these limitations, in this paper, we replace the traditional easy-to-hard paradigm with a beneficial-or-not approach (Mindermann et al., 2022). Instead of relying on predefined difficulty levels, we dynamically curate training samples based on their estimated influence, ensuring that only beneficial samples contribute to model updates. This strategy not only optimizes sample selection but also adapts to the dynamic evolution of sample influence.

**Data scheduling and curation**. A parallel line of work designs policies that reweight or select data during training to improve accuracy per compute. Although conceptually related, the non-influence-based online methods pursue a different objective from our influence-based data valuation framework. Specifically, RHO-Loss and its variants (Mindermann et al., 2022) prioritize samples based on their training difficulty and uncertainty, aiming to accelerate model convergence or optimize resource usage rather than explicitly evaluating sample contributions to validation performance. Similarly, JEST (Evans et al., 2024) focuses on joint multimodal selection to enhance training efficiency, and ACID/ACED (Udandarao et al., 2025) methods target distillation efficiency. GradientIS and GradientMIS (Salaün et al., 2025a;b) adapt the sampling probability of each training example according to the norm of its output-layer gradient, with the primary goal of reducing the variance of the stochastic gradient and thereby accelerating optimization.

However, none of these methods explicitly quantify the per-sample influence on a held-out validation objective, which is central to our research question. Therefore, we position our work as complementary to this parallel line of research, highlighting the distinct goals and evaluation criteria that differentiate our influence-based dynamic valuation approach.

**Online data valuation**. We also acknowledge a few concurrent works (Wang et al., 2024a;b;c) from the same research group that jointly consider online data valuation, emphasizing that sample

gradients play a pivotal role in defining both aspects. However, these approaches face practical challenges, as sample gradients are typically high-dimensional, making them computationally expensive to compute and store (See Section 3). To address these challenges, this paper proposes a layer-aware approximation technique that not only accelerates but also enhances the calculation of sample influence. By leveraging layer-wise structures within deep models, our method reduces computational overhead while maintaining high estimation accuracy for modern deep learning frameworks.

## 3 PRELIMINARIES ON INFLUENCE FUNCTIONS

### 3.1 HESSIAN-FREE INFLUENCE FUNCTIONS

Consider a classifier with parameters $\theta \in \mathbb{R}^D$ mapping instances $z = \{x, y\}$ from input space $x \in \mathcal{X}$ to output space $y \in \mathcal{Y}$, the model parameters $\hat{\theta} = \arg\min_{\theta \in \Theta} \frac{1}{n} \sum_{i=1}^{n} \ell(z_i; \theta)$ can be obtained the empirical risk minimization problem. If we downweight a training sample $z_j$ by a very small fraction $\epsilon$, the new parameters can be $\hat{\theta}(z_j; -\epsilon) = \arg\min_{\theta \in \Theta} \frac{1}{n} (\sum_{i=1}^{n} \ell(z_i; \theta) - \epsilon \ell(z_j; \theta))$. By evaluating the limit as $\epsilon$ approaches 0, the seminal work of Koh and Liang (2017) provides an estimation for the influence score associated with the removal of $z_j$ from the training set:

$$\mathcal{I}(z_j; \hat{\theta}) = -\sum_{z \in \mathcal{V}} \nabla_{\hat{\theta}} \ell(z; \hat{\theta})^\top \mathbf{H}_{\hat{\theta}}^{-1} \nabla_{\hat{\theta}} \ell(z_j; \hat{\theta}), \tag{1}$$

where $\mathcal{V}$ denotes the validation set (self-influence employs the training set instead), $\nabla_{\hat{\theta}} \ell(z_j; \hat{\theta})$ is the gradient of sample $z_j$, and $\mathbf{H}_{\hat{\theta}} = \sum_{i=1}^{n} \nabla_{\hat{\theta}}^2 \ell(z_i; \hat{\theta})$ denotes the Hessian matrix.

Although the above influence functions circumvent the need for model retraining, computing the inverse of the Hessian matrix still poses significant challenges for large deep models. Several approaches have been proposed to approximate the Hessian inverse efficiently (Grosse et al., 2023; Kwon et al., 2023; Park et al., 2023). Recently, there has been a surge of interest in Hessian-free influence functions, which simplify the computation by replacing the Hessian inverse with an identity matrix (Charpiat et al., 2019; Pruthi et al., 2020; Yang et al., 2024). This simplification reduces the influence score calculation to an inner product (IP) between the sample gradients of the validation set and the target sample, formulated as follows:

$$\mathcal{I}^{\text{IP}}(z_j; \hat{\theta}) = -\sum_{z \in \mathcal{V}} \nabla_{\hat{\theta}} \ell(z; \hat{\theta})^\top \cdot \nabla_{\hat{\theta}} \ell(z_j; \hat{\theta}). \tag{2}$$

Since we consider dynamic sample curation for each training batch, the computational cost of evaluating sample impact for each batch remains substantial, especially for large models. Traditional approaches (Koh and Liang, 2017; Grosse et al., 2023; Kwon et al., 2023; Park et al., 2023) that rely on estimating the inverse of the Hessian matrix are prohibitive for dynamic sample impact estimation due to their computational complexity. In this paper, we focus on the IP-based influence score and further propose a layer-aware approximation to enhance the calculation of sample influence.

### 3.2 GHOST OF HESSIAN-FREE INFLUENCE FUNCTIONS

Building on the above IP-based influence approach, Wang et al. (2024c) extended the static influence value to an online version by computing sample influence within each batch. However, this introduces the challenge of frequently calculating sample-level gradients for every batch, which can be computationally expensive. To address this issue, Wang et al. (2024c) proposed the *ghost influence* score, inspired by ghost clipping in differential privacy (Lee and Kifer, 2021). Notably, the inner product of two sample gradients can be decomposed into the product of the inner product between their embeddings and the gradients of the subsequent layer, as shown below:

$$\mathcal{I}^{\text{Ghost}}(z_j; \hat{\theta}) = -\sum_{z \in \mathcal{V}} \sum_{l=1}^{L} \Big( \overbrace{\big((\mathbf{a}_z^{(l-1)})^\top \cdot \mathbf{a}_j^{(l-1)}\big)}^{\alpha^{(l)}} \cdot \overbrace{\big((\frac{\partial \ell^{(l)}}{\partial \mathbf{s}_z^{(l)}})^\top \cdot \frac{\partial \ell^{(l)}}{\partial \mathbf{s}_j^{(l)}}\big)}^{\beta^{(l)}} \Big), \tag{3}$$

where $\mathbf{a}$ and $\mathbf{s}$ are the input/output embeddings, and $l$ is the index of layers. Neglecting the activation function, the above equation can be divided into two parts, $\alpha^{(l)}$ calculates the similarity between a validation sample and the target training sample in the embedding space, and $\beta^{(l)}$ measures the similarity in the gradient space, i.e., the next layer's feedback.

**Limitations of Ghost Influence.** Despite its efficiency gains, ghost influence suffers from two key drawbacks. Computationally, it still requires propagating loss-to-parameter signals through every layer (or materializing parameter-sized, per-sample gradients), which remains costly per batch and difficult to cache at scale. Statistically, mini-batch stochasticity, nonlinear activations/normalization, and residual mixing introduce substantial noise. Because ghost influence sums contributions additively rather than averaging them, this noise can accumulate with depth, making the rankings of hard or noisy examples unstable (see analyses in Appendix A). In this paper, we address both limitations by proposing a layer-aware approximation strategy.

## 4 Methods

In this section, we introduce our layer-aware influence estimator, a simplified approximation of ghost influence. We then analyze its computational and storage costs during training and explain why this lightweight approximation can actually enhance estimation performance.

### 4.1 Layer-aware influence estimator

To address both challenges of ghost influence jointly, we propose a layer-aware influence (LAI) estimator that uses a single, stable feedback channel, while still leveraging multi-layer embeddings. Concretely, we replace all $\beta^{(l)}$ by the last-layer similarity $\beta^{(L)}$ and aggregate embedding similarities across layers:

$$\mathcal{I}^{\text{LAI}}(z_j; \hat{\theta}) = -\sum_{z \in \mathcal{V}} \Big( \sum_{l=1}^{L} ((\mathbf{a}_z^{(l-1)})^\top \cdot \mathbf{a}_j^{(l-1)}) \Big) \cdot \Big( (\frac{\partial \ell^{(L)}}{\partial \mathbf{s}_z^{(L)}})^\top \cdot \frac{\partial \ell^{(L)}}{\partial \mathbf{s}_j^{(L)}} \Big). \tag{4}$$

This design retains the expressive, multi-layer embedding view, yet computes influence using only *output-layer* gradients $\partial \ell^{(L)} / \partial \mathbf{s}^{(L)}$. It eliminates layer-by-layer backpropagation and avoids parameter-sized sample gradients, yielding substantial savings in time and memory. Formally, Eq. (4) is a principled approximation of Eq. (3); the complete derivation is deferred to Appendix B.

### 4.2 Advantages of LAI

**Computational and storage costs.** Compared to the ghost influence in Eq. (3), which requires per-sample feedback at every layer to form $\{\beta^{(l)}\}_{l=1}^{L}$ and thus either materializes intermediate per-sample gradients or runs micro-backward passes, our LAI in Eq. (4) backpropagates only once to the output layer. As a result, scoring a mini-batch against the validation cache scales linearly in both the batch size and $|\mathcal{V}|$ with small constants and does not create parameter-sized per-sample gradients. On the memory side, ghost influence must cache, for each validation point, per-layer gradient signals (or repeatedly recompute them), whose footprint grows with depth and is harder to keep synchronized; in contrast, our method only cache the output-layer gradient signals, which are compact and stable across iterations. In practice, this reduces the backpropagate depth from $L$ to 1 and shrinks the validation cache from $L$ gradient tensors to two short vectors per validation sample, enabling both runtime and storage-efficient online valuation during training.

**Why our simplified approximation improves performance?** At first glance, replacing $\{\beta^{(l)}\}_{l=1}^{L}$ with the single output-layer channel $\beta^{(L)}$ may seem crude. However, as formalized in Appendix A, starts from the output-layer gradients, each per-layer feedback suffers from stochastic perturbations introduced along the backpropagation chain (mini-batch statistics, nonlinear activations/normalization, residual mixing), which aggregates multi-layer noises. By contrast, our LAI employs the single output-layer channel $\beta^{(L)}$ to replace all previous layers, which not only avoids the noise aggregation, but also exhibits lower variance in most common scenarios. A mathematical bias-variance comparison between ghost influence and LAI can be found in Appendix C. Moreover, the superior performance of LAI over ghost influence is empirically validated across diverse experiments in Sections 5 and 6. Taken together, these results show that LAI, as a simplified variant of ghost influence, not only offers substantial computational benefits but also delivers improved performance, which plays a significant advantage of LAI over ghost influence.

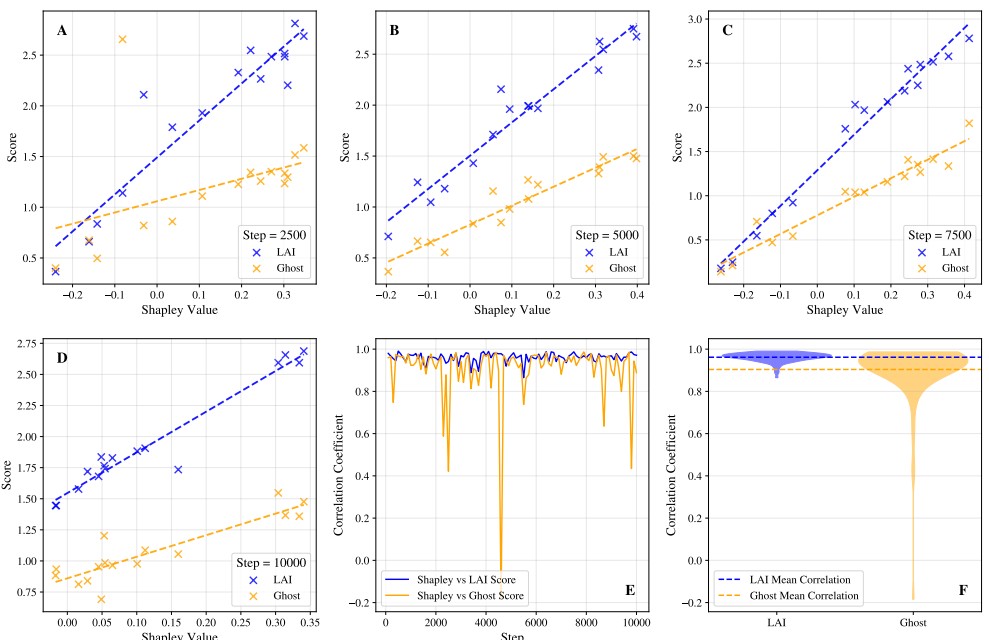

Figure 1: Fidelity Validation of LAI and ghost influence. **A-D** show representative checkpoints at steps 2,500, 5,000, 7,500, and 10,000: each point plots a sample's proxy score (LAI is blue, Ghost is orange) versus its Shapley value at that step, with dashed least-squares fits. **E** reports the per-step Pearson correlation across all 100 checkpoints. **F** summarizes the distribution of these 100 per-step correlations via violin plots with dashed mean lines.

## 5 EXPERIMENTS ON LLMS

In this section, we first conduct fidelity validation of our proposed LAI against a Monte Carlo Shapley reference, and its utility for dynamic batch curation in the scenarios of pre-training and fine-tuning of LLMs, where at each step samples with negative estimated influence are discarded.[1]

### 5.1 FIDELITY VALIDATION

Here we pre-train GPT-Neo (Gao et al., 2020), a 125M-parameter LLM model, for 10,000 iterations, saving checkpoints every 100 steps for a total of 100 checkpoints. At each checkpoint, we compute influence scores over a batch of 16 samples using our LAI, ghost influence, and a Shapley-value reference constructed via 1,000 Monte Carlo permutations (Wang et al., 2024c). We then assess fidelity by measuring Pearson's correlation between each proxy score and the Shapley reference.

In Figure 1, we report four representative snapshots at steps 2,500, 5,000, 7,500, and 10,000 for subfigures **A-D**, and aggregate all 100 checkpoints in subfigures **E** and **F**. Across all the runs, LAI maintains a high and stable fidelity to the Shapley reference (mean = 0.9617, std = 0.0217), whereas Ghost remains positively correlated but fluctuates more (mean = 0.9038, std = 0.1463). Moreover, ghost influence often occurs low correlations at certain steps; for example, ghost influence delivers almost -0.2 correlations with Shapley reference around step 4800. These results demonstrate that our LAI consistently delivers a reliable, high-fidelity estimate of sample influence across the entire pre-training process. This finding confirms that LAI not only serves as an approximation of Ghost influence but also enhances its accuracy and robustness.

### 5.2 LAI FOR PRE-TRAINING LLM

We continue our investigation to assess the feasibility of our LAI for pre-training LLMs. Specifically, we utilize GPT-Neo and further pre-train it using the *Pile-uncopyrighted* dataset. Here we

---

[1]Details on datasets, model training, and experimental setup can be found in AppendixD.

Table 2: Results of fine-tune tasks on *SST2*, *MRPC*, *QNLI*, and *RTE*.

| Dataset | Training Sample Per Epoch | | | | Validation Loss | | | | Test Accuracy (%) | | | |
|---|---|---|---|---|---|---|---|---|---|---|---|---|
| | *SST2* | *MRPC* | *QNLI* | *RTE* | *SST2* | *MRPC* | *QNLI* | *RTE* | *SST2* | *MRPC* | *QNLI* | *RTE* |
| Vanilla | 67,349 | 3,668 | 104,743 | 2,490 | 0.7121 | 1.5022 | 0.9460 | 1.0101 | 89.9 | 72.0 | 83.6 | 59.8 |
| LAI | **34,301** | **1,877** | **52,386** | **833** | **0.5716** | **0.7669** | **0.4978** | **0.6406** | **90.5** | **73.4** | **84.8** | **61.0** |

limit the training process to 10,000 batches for both the baseline method and our LAI. The pre-training performance is subsequently evaluated by perplexity on an unseen corpus from the *Pile-uncopyrighted* dataset, following the evaluation procedure outlined by Gao et al. (2020).

Since the pre-training process lacks a dedicated validation set, we adopt a self-influence approach for batch curation, wherein the current batch samples serve as the validation set to identify and filter out detrimental samples. Specifically, samples with gradient directions opposing the majority within a batch are ex-

Table 1: Results of LLM pre-training tasks on *Pile-uncopyrighted*.

| | Perplexity | | #Removed | Sample |
|---|---|---|---|---|
| Batch | 5000 | 10000 | Sample | Importance |
| Baseline | 8.104929 | 8.086690 | 0 | 1.0000 |
| LAI | **8.104924** | **8.086686** | 135 | **1.0004** |

cluded, which is expected to account for only a small proportion of the data. Table 1 reports the performance comparison between the standard training baseline and our LAI. Using self-influence, only 135 samples are removed across 10,000 batches. Despite this minimal removal rate relative to the vast training dataset and large model parameters, slight improvements are observed. By excluding these identified samples, the importance of each remaining sample in reducing perplexity is enhanced, yielding an average improvement of 0.04% per sample. These results highlight the potential of our LAI method in LLM pre-training, especially with adequate resources.

## 5.3 LAI FOR FINE-TUNING LLM

We further utilize GPT-Neo for fine-tuning evaluation. An additional prediction layer is appended for task-specific fine-tuning. For evaluation, we select four widely used text benchmarks—*SST-2*, *MRPC*, *QNLI*, and *RTE*—from the GLUE repository (Wang et al., 2018), adhering to their official pre-split training, validation, and test sets. During fine-tuning, we optimize both the parameters of the prediction layer and the backbone model using the training set. The standard training baseline is used to warm up the model for 3 epochs, after which we switch to our proposed method. Both methods are fine-tuned for a total of 5 epochs.

Table 2 presents the results of fine-tuning tasks on four benchmark datasets, evaluating sample usage per epoch, validation loss, and test set accuracy. Notably, our method involves processing only half or fewer training samples compared to the standard baseline method. This not only significantly reduces training costs—particularly advantageous for large-scale data and models—but also enhances performance. This observation highlights a crucial insight: not all data contribute positively to learning performance. In fact, undesirable samples can waste computational resources and even degrade learning outcomes. Unfortunately, conventional model training paradigms include all samples in optimization and lack mechanisms to resist the influence of harmful samples. While prior efforts in data curation have aimed to prepare high-quality datasets or remove harmful samples before training, the dynamic nature of sample influence during optimization remains overlooked.

In essence, data valuation is to selective beneficial training samples and exclude detrimental ones, guided by the validation set—specifically, by evaluating whether the samples contribute to reducing validation loss. Consequently, we report the validation loss in these fine-tuning tasks. Across all datasets, our LAI achieves a significant reduction in validation loss. On *SST-2*, validation loss decreases by nearly 20%, while reductions of 30–50% are observed on other three datasets within just two epochs of fine-tuning. This reduction in validation loss directly translates into measurable performance gains in the test sets, with 0.6–1.2% improvements in accuracy.

## 6 EXPERIMENTS ON IMAGE AND TEXT CLASSIFICATION

We continue evaluating our LAI in the scenario of image and text classification. First, we compare our LAI with several static/online data valuation methods and a curriculum learning baseline, then explore the dynamics of sample-level data valuation during the model optimization.

Table 3: Results of classification accuracy on benchmark datasets with noise labels. OOM means out of memory, and OOD means out of disk.

| Methods | CIFAR-10N-a | CIFAR-10N-r | CIFAR-10N-w | CIFAR-100N | 20News-N | Emotion-N | Avg. |
|---|---|---|---|---|---|---|---|
| Vanilla | $85.36_{\pm0.18}$ | $83.19_{\pm0.41}$ | $75.37_{\pm1.05}$ | $40.87_{\pm0.38}$ | $58.99_{\pm0.92}$ | $84.79_{\pm1.55}$ | 71.43 |
| LiSSA (2017) | $78.67_{\pm0.27}$ | $74.15_{\pm0.44}$ | $60.44_{\pm0.52}$ | $10.16_{\pm0.41}$ | $63.41_{\pm0.90}$ | $\mathbf{89.24}_{\pm1.16}$ | 62.68 |
| DataInf (2023) | $79.80_{\pm0.41}$ | $75.34_{\pm0.58}$ | $61.90_{\pm1.53}$ | $30.69_{\pm0.71}$ | $63.36_{\pm0.37}$ | $89.08_{\pm0.51}$ | 66.70 |
| TRAK (2023) | $83.68_{\pm0.25}$ | $80.94_{\pm0.42}$ | $71.77_{\pm0.62}$ | $32.93_{\pm0.83}$ | OOM | OOM | 67.33 |
| IP (2024) | $80.64_{\pm0.48}$ | $78.84_{\pm0.73}$ | $68.28_{\pm0.98}$ | $30.82_{\pm0.86}$ | $\mathbf{63.45}_{\pm0.75}$ | $89.30_{\pm0.54}$ | 68.56 |
| SGD-Inf (2019) | $80.81_{\pm0.00}$ | $81.75_{\pm0.00}$ | $78.40_{\pm0.00}$ | $30.36_{\pm0.00}$ | OOD | OOD | 67.83 |
| MoSo (2023) | $71.74_{\pm0.52}$ | $66.37_{\pm0.70}$ | $38.73_{\pm0.73}$ | $30.02_{\pm0.58}$ | $61.68_{\pm0.98}$ | $86.11_{\pm0.95}$ | 59.11 |
| Z0-Inf (2025) | $81.30_{\pm0.17}$ | $81.30_{\pm0.30}$ | $74.82_{\pm0.40}$ | $37.73_{\pm0.61}$ | $63.41_{\pm0.63}$ | $88.91_{\pm0.57}$ | 72.35 |
| SPL (2010) | $74.60_{\pm2.06}$ | $74.42_{\pm2.46}$ | $65.39_{\pm1.41}$ | $41.96_{\pm0.74}$ | $55.71_{\pm1.33}$ | $79.66_{\pm2.08}$ | 65.29 |
| Ghost (2024c) | $\mathbf{85.52}_{\pm0.31}$ | $83.46_{\pm0.59}$ | $75.98_{\pm0.40}$ | $41.90_{\pm0.63}$ | $63.22_{\pm1.16}$ | $88.36_{\pm0.44}$ | 73.07 |
| LLI (Ours) | $84.13_{\pm0.59}$ | $82.49_{\pm0.16}$ | $75.87_{\pm1.50}$ | $\mathbf{44.22}_{\pm0.56}$ | $63.38_{\pm0.86}$ | $88.64_{\pm0.52}$ | 73.12 |
| LAI (Ours) | $84.78_{\pm0.52}$ | $\mathbf{83.69}_{\pm0.11}$ | $\mathbf{76.43}_{\pm0.39}$ | $44.06_{\pm0.73}$ | $63.23_{\pm0.97}$ | $88.45_{\pm0.54}$ | $\mathbf{73.44}$ |

## 6.1 Algorithmic Performance

We evaluate three categories of competitive methods: (1) static influence-based two-round training approaches, (2) online one-round training approaches, and (3) the curriculum learning baseline. For the static influence-based methods, we include the well-known LiSSA (Koh and Liang, 2017), DataInf (Kwon et al., 2023), TRAK (Park et al., 2023), IP (Yang et al., 2024), and three non-convex, optimization-aware influence baselines SGD-Influence (Hara et al., 2019), MoSo (Tan et al., 2023), and Z0-Inf (Kokhlikyan et al., 2025). For the online methods, we include Ghost Influence (Wang et al., 2024c), our proposed Layer-Aware Influence (LAI), and Layer-aware Last-layer Influence (LLI)—a variant of LAI that relies solely on the last-layer embedding and gradient for influence computation. As the curriculum learning baseline, we consider Self-Paced Learning (SPL) (Kumar et al., 2010) as our baseline for comparison.

Table 3 presents the average classification accuracy and standard deviation of all competitive methods over five runs. In image datasets, static influence-based two-round training methods often result in extended training time and inferior performance compared to the vanilla approach. This inefficacy arises from the inherent limitations of static influence estimation: samples deemed detrimental in a converged model may no longer have the same impact during subsequent training iterations. This issue becomes especially severe in non-convex optimization settings, particularly when a large number of samples are removed. These methods primarily focus on model-associated sample influence derived from a fixed model, while they overlook the full dynamic evolution of influence that occurs throughout optimization process, ultimately limiting their overall effectiveness in practice.

In contrast, on text datasets, static influence-based methods tend to perform effectively by successfully removing identified detrimental samples. This disparity likely stems from differences in training procedures: image datasets typically involve training the backbone architecture from scratch, while text datasets leverage pre-trained backbones for fine-tuning, thereby retaining a significant portion of the original model's knowledge and mitigating the impact of removed samples. An intriguing observation is that IP, a remarkably simple and naive method that avoids using the Hessian matrix, outperforms other static influence-based methods employing more sophisticated Hessian approximations. This result reinforces the rationale behind our choice to build online data valuation framework on such a straightforward method, highlighting its simplicity, efficiency, and effectiveness. Conversely, TRAK, despite its innovative approach, demands significantly more computational resources and often encounters out-of-memory issues in our experimental environment.

For non-convex, optimization-aware influence baselines, on *CIFAR-10N*, SGD-Influence attains accuracy comparable to IP and TRAK, and gives strong results on *CIFAR-10N-w*. However, it requires storing all training checkpoints, which makes it impractical for the larger text models: on *20News-N* and *Emotion-N* it would need roughly 0.4 TB and 0.6 TB of disk space, respectively, exceeding the capacity of our workstation, and therefore appears as out-of-disk. MoSo performs reasonably well on the text datasets but is much weaker than most methods on the noisy *CIFAR-10N* variants, suggesting that its pruning strategy is brittle under heavy label noise when training vision models from scratch. Z0-Inf achieves the best average accuracy among these three baselines, clearly improving over earlier static influence methods such as LiSSA, DataInf, and IP, yet it still lags behind our online

Table 4: Time and memory comparison of online data valuation approaches.

| Dataset | GFLOP per Batch | | | | Maximum Memory | | | |
|---|---|---|---|---|---|---|---|---|
| | *CIFAR-10N* | *CIFAR-100N* | *20News-N* | *Emotion-N* | *CIFAR-10N* | *CIFAR-100N* | *20News-N* | *Emotion-N* |
| Vanilla | 57.24 | 57.32 | 1073.91 | 1073.90 | 3304.67 | 3305.49 | 5292.08 | 5291.83 |
| Ghost | 209.13 | 209.32 | 2156.07 | 2872.02 | 4223.74 | 4227.10 | 12013.19 | 12013.38 |
| LLI | 113.15 | 113.34 | 835.26 | 1551.20 | 3328.60 | 3330.28 | 7552.19 | 7552.37 |
| LAI | 113.42 | 113.61 | 839.49 | 1555.27 | 3716.95 | 3718.64 | 8053.50 | 9053.64 |

methods LLI and LAI and requires extra passes over the data to estimate zeroth-order influences. Overall, these additional baselines indicate that optimization-aware static influence methods alone are not sufficient to match the performance and practicality of online data valuation in our setting.

While curriculum learning approaches like SPL incorporate online data considerations, they show inconsistent results across datasets, confirming prior findings (Kumar et al., 2010; Zaremba and Sutskever, 2014; Hacohen and Weinshall, 2019) that loss-based curricula do not consistently yield performance gains. In contrast, LLI and LAI dynamically integrate data valuation, delivering superior performance to the vanilla method. It is worth noting that ghost influence can be viewed as the dynamic version of IP, achieving over 5% improvement on average, which underscores the effectiveness of the online data valuation framework. Notably, our LAI further outperforms ghost influence in most cases, particularly on the challenging *CIFAR-100N* dataset. This superior performance is achieved while requiring fewer computational resources—a critical advantage that will be further discussed in the next paragraph. Intuitively, a simplified method often trades performance for lower computational cost. However, the superior results of LAI show that, while it is designed as a simplified version of ghost influence for efficiency, it not only avoids sacrificing accuracy but also delivers an additional performance gain.

We further investigate the computational cost of online methods in terms of running time and memory usage, as presented in Table 4. We report the runtime of the vanilla method as a reference point, but our main objective is to compare methods within the category of online data valuation, rather than contrasting online valuation with the vanilla baseline. Note that static data valuation methods require two rounds of training—roughly doubling the runtime of the vanilla method—plus additional time for data valuation computation; therefore, we do not report them here. A significant limitation of ghost influence lies in its reliance on the complete sample gradient to calculate sample influence. For instance, ghost influence takes over 2.5 times running time and nearly 1.5 times memory over LLI or LAI on *20News-N* in terms of GFLOP, which might be further amplified on large networks. While the pairwise strategy helps avoid setting the batch size to one for sample-level gradient computation, the requirement for complete sample gradients leads to considerable computational and storage demands. The sample gradient's dimensionality matches that of the model parameters, resulting in high costs for gradient computation for each sample in every batch. Additionally, the large storage requirements restrict ghost influence to be conducted on small batch size. These challenges highlight the need for more efficient gradient estimation methods, which are well addressed by our LAI. Our LAI utilizes only the output gradient, combined with embeddings from each layer, offering a more memory-efficient and computationally manageable approach. Notably, LAI requires almost the same computational resources as LLI, while maintaining improved performance in general. Besides, while all the embeddings are used in LAI and only the last layer embedding is used in LLI, the last layer embedding also requires the calculation of previous embeddings as inputs. Therefore, they takes similar computational resources.

## 6.2 EXPLORATION ON LAI AT SAMPLE LEVEL

Beyond evaluating overall algorithmic performance, we delve deeper into the sample influence dynamics of our LAI framework, as depicted in Figure 2. Subfigure **A** illustrates the collective distribution of influence scores for training samples across epochs. Over time, these scores converge sharply around zero, signifying reduced variability in sample influence. Samples with substantial negative influence, which degrade validation loss, are excluded from subsequent epochs, while those with negligible influence exert minimal impact on model parameters. By setting the influence threshold to zero, approximately half of the samples are retained for optimization at each epoch.

At the individual level, Subfigure **B** visualizes sample involvement across epochs, with blue indicating inclusion in a batch and white indicating exclusion. Generally, very few "easy" or "hard"

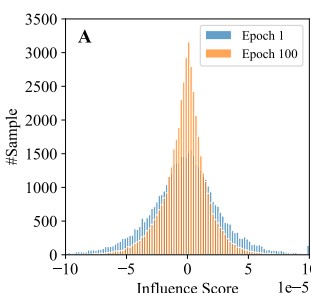 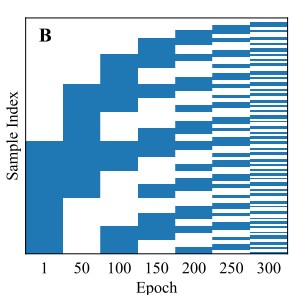 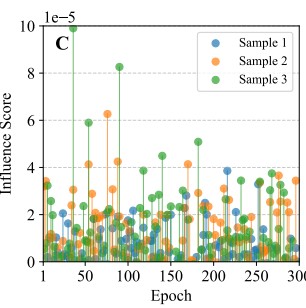

Figure 2: Sample influence dynamics of LAI on *CIFAR10-N-w*. **A** illustrates the collective distribution of influence scores across different epochs; **B** visualizes sample involvement across epochs, with blue indicating inclusion in a batch and white indicating exclusion; **C** presents dynamic nature by tracing the influence scores of three specific samples across epoch.

samples consistently drop out early or join late in training, which may explain the limitations of traditional curriculum learning. Instead, the composition of batches evolves dynamically, highlighting the importance of online data valuation. Subfigure **C** examines this dynamic nature by tracing the influence scores of three specific samples across epochs. Influence scores shift significantly, as samples used in one epoch often exhibit reduced loss in subsequent epochs. However, these samples may no longer contribute maximally to reducing loss in future epochs, justifying their exclusion. This also well justifies the limitation of static data valuation.

# 7    LIMITATION AND FUTURE DIRECTION

Despite these encouraging results, our study has several limitations. First, our method follows the classical influence-function derivation, which assumes local convexity, smoothness, and the existence of a stationary point. Directly applying this theory to non-convex deep networks is not formally justified, although this is a common practice in recent influence-function work on deep models. Second, our specific gradient decompositions and noise-propagation analysis, as well as the Ghost influence baseline (Wang et al., 2024c) we build on, are derived under simplified network models that, like the Ghost influence formulation, focus on the linear transformation components of each layer. This covers most standard layers in deep networks (such as fully connected and convolutional layers), but inherently non-linear or stateful components—for example normalization layers and certain regularization blocks—are not modeled, so the per-layer interpretation we provide is only an approximate description of modern architectures.

Future work on online data valuation includes developing more efficient and scalable influence estimation techniques that can operate in streaming or continually evolving datasets. Another promising direction is integrating online valuation with adaptive sampling or curriculum-learning strategies, allowing models to prioritize the most informative or influential examples during training. Additionally, establishing theoretical guarantees for stability, convergence, and robustness in online valuation frameworks remains an important challenge. Together, these avenues can advance more principled and effective data-centric learning systems.

# 8    CONCLUSION

In this paper, we tackled a fundamental challenge from a data-centric perspective by introducing the online data valuation framework. This framework integrates online data valuation to enhance model optimization while providing an efficient and generic implementation compatible with SGD and Adam optimization. Specifically, we utilized a Hessian-free influence function to evaluate the quality of samples within each batch, dynamically removing detrimental samples from the optimization process. To address the computational overhead of frequent sample influence estimation, we developed an efficient layer-aware approximation to streamline the calculation. Extensive experiments validated the effectiveness and efficiency of our approach by comparing with other baseline methods across diverse scenarios, including LLM pre-training/fine-tuning and image/text classification.

## REPRODUCIBILITY STATEMENT

We provide our code, instructions, and implementation in an open-source repository: `https://anonymous.4open.science/r/Dynamic-Batch-Curation-8782`. The experiments in Section 5 were conducted on Google Cloud TPU v4-4 nodes (Ubuntu 22.04.2 LTS) with PyTorch. The experiments in Section 6 were conducted on a Linux (Ubuntu 20.04.6 LTS) server using NVIDIA GeForce RTX 4090 GPUs with 24GB VRAM running CUDA version 12.3 and driver version 545.23.08.

## ETHICS STATEMENT

Our method introduces online data valuation with a layer-aware influence estimator (LAI) that filters harmful samples on the fly, reducing compute and energy while improving generalization without extra training epochs. Nevertheless, potential risks exist: 1) amplification of validation-set bias, 2) possible exclusion of long-tail groups, and 3) privacy concerns from caching embeddings and last-layer gradients. To mitigate these, we recommend careful design of the validation set, subgroup-aware evaluation, and secure handling of cached information. Finally, while LAI aligns most closely with SGD by leveraging last-layer gradients, when using adaptive optimizers such as Adam there may be mild direction mismatches. We provide a lightweight remedy and guidance in Appendix E.

## ACKNOWLEDGMENT

We sincerely thank the authors of the baseline methods for making their code publicly available, which greatly facilitated our implementation and comparison. We also appreciate the valuable feedback and constructive suggestions provided by the reviewers, which have helped us significantly improve the clarity, rigor, and overall quality of this work.

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

APPENDIX

Appendix A–C provide the theoretical analyses of ghost influence, our LAI estimator, and their relationship. Specifically, we first characterize how ghost influence propagates along the backpropagation chain, then formalize LAI as a structured approximation to ghost influence, and finally compare their respective bias–variance tradeoffs. Together, these analyses explain why the simplified layer-aware estimator achieves more stable and accurate data valuation. Appendix D provides complementary details on the datasets and models, followed by Appendix E, which offers additional practical guidance.

## A  NOISE PROPAGATION ANALYSIS OF GHOST

This section formalizes the statistical picture behind Eq. (3). We proceed progressively: (i) identify a low-noise anchor at the output layer; (ii) explain how depth injects and transforms noise through random linear backpropagation operators; (iii) project vector-level perturbations to the per-layer scalar similarity $\beta^{(l)}$; and (iv) quantify depth-wise accumulation under the ghost influence. All symbols follow the main text.

For a sample $x \in \{z, j\}$, where $z$ is a validation sample and $j$ is the target training sample, and layer $l$, we define

$$\mathbf{g}_x^{(l)} := \frac{\partial \ell^{(l)}}{\partial \mathbf{s}_x^{(l)}} \in \mathbb{R}^{d_l}, \qquad \beta_{z,j}^{(l)} := \left(\mathbf{g}_z^{(l)}\right)^\top \mathbf{g}_j^{(l)}, \qquad \beta_{z,j}^\star := \left(\mathbf{g}_z^{(L)}\right)^\top \mathbf{g}_j^{(L)}.$$

### A.1  A LOW-NOISE ANCHOR: THE OUTPUT-LAYER FEEDBACK $\beta^{(L)} = \beta^\star$

We first define $\beta_{v,j}^\star$, the shared signal between validation and training gradients, which is the common layer-invariant component reflecting the alignment. Since the last-layer backpropagation operator is the identity, the output-layer similarity equals the shared signal:

$$\beta_{z,j}^{(L)} = \beta_{z,j}^\star.$$

This motivates using the output-layer channel as a stable anchor in our estimator. Corresponding analyses for other layers can be found in the following Section A.3.

### A.2  HOW DEPTH INJECTS NOISE: RANDOM LINEAR BACKPROPAGATION OPERATORS

Backpropagation from layer $l$ to $l - 1$ is linear in the upstream gradient and can be written as a vector–Jacobian product:

$$\mathbf{g}_x^{(l-1)} = J_x^{(l)} \, \mathbf{g}_x^{(l)}, \qquad l = 1, \ldots, L.$$

We decompose the sample-dependent backpropagation operator as $J_x^{(l)} = \overline{J}^{(l)} + \Delta J_x^{(l)}$, where $\overline{J}^{(l)}$ is the systematic Jacobian (e.g., population/EMA statistics for normalization) and $\Delta J_x^{(l)}$ captures stochasticity induced by mini-batch statistics, activation gating, and dropout masks (Faghri et al., 2020; Srivastava et al., 2014; Santurkar et al., 2018). No additive constant independent of the upstream gradient is introduced. At the output layer, we allow fluctuations caused by the forward pass as follows:

$$\mathbf{g}_x^{(L)} = \mathbf{g}_x^{\star(L)} + \xi_x^{(L)}, \qquad \mathbb{E}[\xi_x^{(L)}] = \mathbf{0}.$$

### A.3  FROM VECTORS TO SCALARS: PER-LAYER SIMILARITY AND ITS DECOMPOSITION

Composing the layerwise maps yields the depth-$l$ backpropagation operator

$$\mathbf{g}_x^{(l)} = A_x^{(l)} \, \mathbf{g}_x^{(L)}, \qquad A_x^{(l)} := J_x^{(l+1)} J_x^{(l+2)} \cdots J_x^{(L)}. \tag{5}$$

Let $\overline{A}^{(l)} := \overline{J}^{(l+1)} \cdots \overline{J}^{(L)}$ and write $A_x^{(l)} = \overline{A}^{(l)} + \Delta A_x^{(l)}$. A first-order expansion of the product gives

$$\Delta A_x^{(l)} \approx \sum_{t=l+1}^{L} \left(\overline{J}^{(l+1)} \cdots \overline{J}^{(t-1)}\right) \Delta J_x^{(t)} \left(\overline{J}^{(t+1)} \cdots \overline{J}^{(L)}\right), \tag{6}$$

with higher-order products of $\{\Delta J_x^{(t)}\}$ absorbed into the residual. Using $\mathbf{g}_x^{(L)} = \mathbf{g}_x^{\star(L)} + \xi_x^{(L)}$,

$$\mathbf{g}_x^{(l)} = \overline{A}^{(l)} \mathbf{g}_x^{\star(L)} + \overline{A}^{(l)} \xi_x^{(L)} + \Delta A_x^{(l)} \mathbf{g}_x^{\star(L)} + \Delta A_x^{(l)} \xi_x^{(L)}. \tag{7}$$

The per-layer scalar similarity in Eq. (3) can be written as

$$\beta_{z,j}^{(l)} = \left(\mathbf{g}_z^{(l)}\right)^\top \mathbf{g}_j^{(l)} = \left(\mathbf{g}_z^{(L)}\right)^\top M_{z,j}^{(l)} \mathbf{g}_j^{(L)}, \qquad M_{z,j}^{(l)} := \left(A_z^{(l)}\right)^\top A_j^{(l)}. \tag{8}$$

Define $\overline{M}^{(l)} := (\overline{A}^{(l)})^\top \overline{A}^{(l)}$ and decompose it into an isotropic gain and an anisotropic remainder:

$$c_l := \frac{1}{d_l} \operatorname{tr} \overline{M}^{(l)}, \qquad \overline{E}^{(l)} := \overline{M}^{(l)} - c_l I.$$

Substituting $A_x^{(l)} = \overline{A}^{(l)} + \Delta A_x^{(l)}$ into Eq. (8) and grouping terms yields

$$\beta_{z,j}^{(l)} = c_l \beta_{z,j}^\star + \varepsilon_{z,j}^{(l)}, \tag{9}$$

where the scalar noise

$$\varepsilon_{z,j}^{(l)} = \left(\mathbf{g}_z^{(L)}\right)^\top \overline{E}^{(l)} \mathbf{g}_j^{(L)} + \left(\mathbf{g}_z^{(L)}\right)^\top \left(\overline{A}^{(l)}\right)^\top \Delta A_j^{(l)} \mathbf{g}_j^{(L)} + \left(\mathbf{g}_z^{(L)}\right)^\top \left(\Delta A_z^{(l)}\right)^\top \overline{A}^{(l)} \mathbf{g}_j^{(L)}$$
$$+ \left(\mathbf{g}_z^{(L)}\right)^\top \left(\Delta A_z^{(l)}\right)^\top \Delta A_j^{(l)} \mathbf{g}_j^{(L)}$$
$$+ \text{ terms involving } \xi_z^{(L)} \text{ and } \xi_j^{(L)} \text{ from Eq. (7).} \tag{10}$$

For $l = L$ we have $A_x^{(L)} = I$, hence $M_{z,j}^{(L)} = I$, $c_L = 1$, and $\overline{E}^{(L)} = 0$. Therefore

$$\beta_{z,j}^{(L)} = \beta_{z,j}^\star \quad \text{exactly.} \tag{11}$$

### A.4 Depth-wise accumulation under ghost influence

Let $\rho_{u \to v} := \prod_{t=u}^v \rho_t$ with $\rho_{u \to v} = 1$ if $u > v$. From Eq. (6), we have

$$\|\Delta A_x^{(l)}\| \lesssim \sum_{t=l+1}^L \rho_{l+1 \to t-1} \|\Delta J_x^{(t)}\| \rho_{t+1 \to L}, \tag{12}$$

ignoring higher-order perturbation products. Combining Eqs. (9)–(10) with Eq. (12) yields the schematic bound

$$\operatorname{Var}\left[\beta_{z,j}^{(l)}\right]$$
$$\lesssim c_l^2 \operatorname{Var}\left[\beta_{z,j}^\star\right] + \kappa_l \|\mathbf{g}_z^{(L)}\|^2 \|\mathbf{g}_j^{(L)}\|^2 \left(\|\overline{E}^{(l)}\|_{\mathrm{F}}^2 + \sum_{t=l+1}^L \rho_{l+1 \to t-1}^2 \rho_{t+1 \to L}^2 \mathbb{E}\|\Delta J^{(t)}\|_{\mathrm{F}}^2\right)$$
$$+ \text{ cross terms,} \tag{13}$$

where $\kappa_l$ depends on norms of $\overline{A}^{(l)}$, and the cross terms collect covariances across layers and between $\Delta A^{(l)}$ and the output-layer fluctuations $\xi^{(L)}$. For the ghost influence, $\mathcal{I}^{\mathrm{Ghost}}(z_j; \hat{\theta}) := -\sum_{z \in \mathcal{V}} \sum_{l=1}^L \alpha_{z,j}^{(l)} \beta_{z,j}^{(l)}$, the corresponding variance can be written as follows:

$$\operatorname{Var}\left[\sum_{l=1}^L \alpha_{z,j}^{(l)} \beta_{z,j}^{(l)}\right] = \left(\sum_{l=1}^L \alpha_{z,j}^{(l)} c_l\right)^2 \operatorname{Var}\left[\beta_{z,j}^\star\right] + \sum_{l=1}^L \left(\alpha_{z,j}^{(l)}\right)^2 \operatorname{Var}\left[\varepsilon_{z,j}^{(l)}\right]$$
$$+ 2 \sum_{1 \le l < k \le L} \alpha_{z,j}^{(l)} \alpha_{z,j}^{(k)} \operatorname{Cov}\left[\varepsilon_{z,j}^{(l)}, \varepsilon_{z,j}^{(k)}\right]. \tag{14}$$

When many cross-layer covariances are nonnegative, the variance grows faster than linearly with depth, and cross-layer sign/scale inconsistencies can induce cancellations in the aggregated score.

## A.5 REMARKS AND CONNECTION TO THE LIMITATIONS PARAGRAPH

The decomposition $\beta^{(l)} = c_l \beta^\star + \varepsilon^{(l)}$ arises from vector-level perturbations through the bilinear form Eq. (9), and the additive aggregation $\sum_l \alpha^{(l)} \varepsilon^{(l)}$ explains why Ghost accumulates depth-wise noise and suffers from cross-layer cancellations. Using the single output-layer channel $\beta^{(L)} = \beta^\star$ avoids these issues while keeping the multi-layer embedding similarities $\sum_l \alpha^{(l)}$, which underpins the limitations highlighted in the main text.

## B  LAI IS AN APPROXIMATION OF GHOST INFLUENCE

To make it formal, we define embedding and gradient similarities between a validation sample $z$ and a target training sample $j$ at the $l$-level as follows:

$$\alpha_{z,j}^{(l)} := \langle \mathbf{a}_z^{(l-1)}, \mathbf{a}_j^{(l-1)} \rangle, \qquad \beta_{z,j}^{(l)} := \langle \mathbf{g}_z^{(l)}, \mathbf{g}_j^{(l)} \rangle.$$

The ghost influence and layer-aware (LAI) influence scores are

$$\mathcal{I}^{\text{Ghost}}(z_j; \hat{\theta}) := -\sum_{z \in \mathcal{V}} \sum_{l=1}^{L} \alpha_{z,j}^{(l)} \beta_{z,j}^{(l)}, \qquad \mathcal{I}^{\text{LAI}}(z_j; \hat{\theta}) := -\sum_{z \in \mathcal{V}} \Big( \sum_{l=1}^{L} \alpha_{z,j}^{(l)} \Big) \beta_{z,j}^{(L)}. \tag{15}$$

To formally establish that LAI serves as an approximation of ghost influence, we first introduce a set of mild assumptions commonly adopted in theoretical deep learning analysis. Under these assumptions, we derive a closed-form expression for the difference between ghost influence and LAI and prove that this difference is upper-bounded by a constant, thereby demonstrating the theoretical soundness of LAI as a proxy.

Here are the assumptions we use.

(A1) *Gradient-norm decay.* There exists $\rho \in (0, 1)$ such that $\|\mathbf{g}_p^{(l)}\|_2 \leq \rho^{L-l} \|\mathbf{g}_p^{(L)}\|_2$ for all $p \in \{z, j\}$ and $l = 1, \ldots, L$ (*cf.* vanishing/residual-gradient behavior (Glorot and Bengio, 2010; He et al., 2016)).

(A2) *Bounded activations.* There exists $C_a > 0$ with $\|\mathbf{a}_p^{(l)}\|_2 \leq C_a$ for all $p$ and $l$ (encouraged by normalization and Lipschitz activations (Ioffe and Szegedy, 2015)), hence $|\alpha_{z,j}^{(l)}| \leq C_a^2$.

(A3) *Non-negative gradient alignment.* $\cos\big(\mathbf{g}_z^{(l)}, \mathbf{g}_j^{(l)}\big) \geq 0$ for all $l$.

The difference between ghost influence and LAI can be written as follows:

$$\zeta_{z,j} := \sum_{l=1}^{L-1} \alpha_{z,j}^{(l)} \big( \beta_{z,j}^{(l)} - \beta_{z,j}^{(L)} \big), \qquad \mathcal{I}^{\text{Ghost}}(z_j; \hat{\theta}) - \mathcal{I}^{\text{LAI}}(z_j; \hat{\theta}) = -\sum_{z \in \mathcal{V}} \zeta_{z,j}. \tag{16}$$

**Conservative bound under (A1)–(A3).** Using $|\alpha_{z,j}^{(l)}| \leq C_a^2$ and $\beta_{z,j}^{(l)} \geq 0$,

$$\big| \zeta z, j \big| \leq C_a^2 \sum_{l=1}^{L-1} \big( \beta_{z,j}^{(L)} + \beta_{z,j}^{(l)} \big).$$

By (A1) and Cauchy–Schwarz, $\beta_{z,j}^{(l)} \leq \|\mathbf{g}_z^{(l)}\|_2 \|\mathbf{g}_j^{(l)}\|_2 \leq \rho^{2(L-l)} \|\mathbf{g}_z^{(L)}\|_2 \|\mathbf{g}_j^{(L)}\|_2$. Therefore

$$\big| \mathcal{I}^{\text{Ghost}}(z_j; \hat{\theta}) - \mathcal{I}^{\text{LAI}}(z_j; \hat{\theta}) \big| \leq C_a^2 \sum_{z \in \mathcal{V}} \Big[ (L-1) \beta_{z,j}^{(L)} + \|\mathbf{g}_z^{(L)}\|_2 \|\mathbf{g}_j^{(L)}\|_2 \sum_{l=1}^{L-1} \rho^{2l} \Big]. \tag{17}$$

Moreover, since $\sum_{l=1}^{L} \alpha_{z,j}^{(l)} \geq \alpha_{z,j}^{(L)} \geq \bar{\alpha}$ and $\beta_{z,j}^{(L)} \geq 0$,

$$\frac{\big| \mathcal{I}^{\text{Ghost}}(z_j; \hat{\theta}) - \mathcal{I}^{\text{LAI}}(z_j; \hat{\theta}) \big|}{\big| \mathcal{I}^{\text{LAI}}(z_j; \hat{\theta}) \big|} \leq \frac{C_a^2}{\bar{\alpha}} \left[ (L-1) + \frac{\sum_z \|\mathbf{g}_z^{(L)}\|_2 \|\mathbf{g}_j^{(L)}\|_2}{\sum_z \beta_{z,j}^{(L)}} \cdot \frac{\rho^2 \big( 1 - \rho^{2(L-1)} \big)}{1 - \rho^2} \right]. \tag{18}$$

**Geometric relative error under a non-expansive alignment condition.** Empirically one often observes that alignment does not increase when backpropagating to lower layers, which leads the variant of A3 that

$$\cos\big(\mathbf{g}_z^{(l)}, \mathbf{g}_j^{(l)}\big) \ \leq \ \cos\big(\mathbf{g}_z^{(L)}, \mathbf{g}_j^{(L)}\big) \quad \forall l \leq L.$$

Then $\beta_{z,j}^{(l)} \leq \rho^{\,2(L-l)}\,\beta_{z,j}^{(L)}$, and hence

$$\big|\mathcal{I}^{\text{Ghost}}(z_j;\hat{\theta}) - \mathcal{I}^{\text{LAI}}(z_j;\hat{\theta})\big| \ \leq \ C_a^2\bigg(\sum_{z\in\mathcal{V}} \beta_{z,j}^{(L)}\bigg) \sum_{l=1}^{L-1} \rho^{\,2l} \ = \ C_a^2\bigg(\sum_{z\in\mathcal{V}} \beta_{z,j}^{(L)}\bigg) \frac{\rho^2\big(1 - \rho^{2(L-1)}\big)}{1 - \rho^2}. \tag{19}$$

Dividing by $|\mathcal{I}^{\text{LAI}}(z_j;\hat{\theta})| \geq \bar{\alpha}\sum_z \beta_{z,j}^{(L)}$ yields

$$\frac{\big|\mathcal{I}^{\text{Ghost}}(z_j;\hat{\theta}) - \mathcal{I}^{\text{LAI}}(z_j;\hat{\theta})\big|}{\big|\mathcal{I}^{\text{LAI}}(z_j;\hat{\theta})\big|} \ \leq \ \frac{C_a^2}{\bar{\alpha}} \cdot \frac{\rho^2\big(1 - \rho^{2(L-1)}\big)}{1 - \rho^2} \ = \ \mathcal{O}(\rho^2) \quad (\rho \to 0). \tag{20}$$

Thus the relative error decays geometrically with depth or smaller $\rho$, while LAI discards $L-1$ layers of per-sample feedback, reducing memory/FLOPs by roughly an order of magnitude.

## C  BIAS–VARIANCE COMPARISON BETWEEN GHOST INFLUENCE AND LAI

Here we analyze the difference between Ghost Influence and LAI from the bias-variance perspective, and demonstrate why LAI, a simplified approximation, is even better than ghost influence.

Following the previous notations,

$$\mathcal{I}^{\text{Ghost}}(z_j;\hat{\theta}) = -\sum_{l=1}^{L} \alpha_{z,j}^{(l)}\big(c_l\,\beta_{z,j}^\star + \varepsilon_{z,j}^{(l)}\big) = -\bigg(\sum_{l=1}^{L} \alpha_{z,j}^{(l)}c_l\bigg)\beta_{z,j}^\star \ - \ \sum_{l=1}^{L} \alpha_{z,j}^{(l)}\varepsilon_{z,j}^{(l)}.$$

$$\mathcal{I}^{\text{LAI}}(z_j;\hat{\theta}) = -\bigg(\sum_{l=1}^{L} \alpha_{z,j}^{(l)}\bigg)\beta_{z,j}^{(L)} = -\bigg(\sum_{l=1}^{L} \alpha_{z,j}^{(l)}\bigg)\beta_{z,j}^\star,$$

Here we further define two extra variables below to better decomposite the ghost influence

$$X \ := \ \bigg(\sum_{l=1}^{L} \alpha_{z,j}^{(l)}c_l\bigg)\beta_{z,j}^\star, \qquad Y \ := \ \sum_{l=1}^{L} \alpha_{z,j}^{(l)}\varepsilon_{z,j}^{(l)}.$$

Then we have the variance of ghost influence

$$\text{Var}\Big[\mathcal{I}^{\text{Ghost}}(z_j;\hat{\theta})\Big] = \text{Var}[X + Y] \tag{21}$$

$$= \bigg(\sum_{l=1}^{L} \alpha_{z,j}^{(l)}c_l\bigg)^2 \text{Var}\big[\beta_{z,j}^\star\big] + \text{Var}[Y] + 2\bigg(\sum_{l=1}^{L} \alpha_{z,j}^{(l)}c_l\bigg)\text{Cov}\big(\beta_{z,j}^\star,\, Y\big). \tag{22}$$

Moreover, expanding the noise term yields

$$\text{Var}[Y] = \sum_{l=1}^{L} \big(\alpha_{z,j}^{(l)}\big)^2 \text{Var}\big[\varepsilon_{z,j}^{(l)}\big] + 2 \sum_{1\leq l < k \leq L} \alpha_{z,j}^{(l)}\alpha_{z,j}^{(k)}\,\text{Cov}\big[\varepsilon_{z,j}^{(l)},\, \varepsilon_{z,j}^{(k)}\big]. \tag{23}$$

According to Cauchy–Schwarz lower bound, for any random variables $X, Y$,

$$\text{Var}[X+Y] \ \geq \ \Big(\sqrt{\text{Var}[X]} - \sqrt{\text{Var}[Y]}\Big)^2.$$

Applying this with the above $X$ and $Y$ gives the unconditional bound

$$\text{Var}\Big[\mathcal{I}^{\text{Ghost}}(z_j;\hat{\theta})\Big] \ \geq \ \Big(\big|\textstyle\sum_{l=1}^{L} \alpha_{z,j}^{(l)}c_l\big|\sqrt{\text{Var}[\beta_{z,j}^\star]} \ - \ \sqrt{\text{Var}[Y]}\Big)^2. \tag{24}$$

Given $\mathrm{Cov}\bigl(\beta_{z,j}^{\star}, Y\bigr) \geq 0$ and $\mathrm{Cov}\bigl(\varepsilon_{z,j}^{(l)}, \varepsilon_{z,j}^{(k)}\bigr) \geq 0$ for $l \neq k$, combining Eq. (21) and (23) yields the stronger lower bound

$$
\begin{aligned}
&\mathrm{Var}\Bigl[\mathcal{I}^{\mathrm{Ghost}}(z_j; \hat{\theta})\Bigr] \\
&\geq \Bigl(\sum_{l=1}^{L} \alpha_{z,j}^{(l)} c_l\Bigr)^2 \mathrm{Var}[\beta_{z,j}^{\star}] + \sum_{l=1}^{L} \bigl(\alpha_{z,j}^{(l)}\bigr)^2 \mathrm{Var}[\varepsilon_{z,j}^{(l)}] + 2 \sum_{1 \leq l < k \leq L} \alpha_{z,j}^{(l)} \alpha_{z,j}^{(k)} \mathrm{Cov}[\varepsilon_{z,j}^{(l)}, \varepsilon_{z,j}^{(k)}].
\end{aligned}
\tag{25}
$$

In particular, dropping the signal term yields the noise-only bound $\mathrm{Var}[\mathcal{I}^{\mathrm{Ghost}}(z_j; \hat{\theta})] \geq \mathrm{Var}[Y] = \sum_l (\alpha_{z,j}^{(l)})^2 \mathrm{Var}[\varepsilon_{z,j}^{(l)}] + 2 \sum_{l<k} \alpha_{z,j}^{(l)} \alpha_{z,j}^{(k)} \mathrm{Cov}[\varepsilon_{z,j}^{(l)}, \varepsilon_{z,j}^{(k)}]$.

By contrast, using $\beta^{(L)} = \beta^{\star}$,

$$
\mathrm{Var}\bigl[\mathcal{I}^{\mathrm{LAI}}(z_j; \hat{\theta})\bigr] = \Bigl(\sum_{l=1}^{L} \alpha_{z,j}^{(l)}\Bigr)^2 \mathrm{Var}[\beta_{z,j}^{\star}].
\tag{26}
$$

With $c_l \approx 1$ (otherwise the average backpropagation gain deviates too much from isometry, making gradients explode/vanish and training impractical), Ghost and LAI share the same signal scaling. The key difference is variance: LAI keeps only the output-layer channel, yielding $\mathrm{Var}[\mathcal{I}^{\mathrm{LAI}}] = (\sum_l \alpha_{z,j}^{(l)})^2 \mathrm{Var}[\beta_{z,j}^{\star}]$, whereas Ghost additionally aggregates the noise $Y$ and cross-layer covariances. These extra terms inflate variance when correlations are nonnegative; even without such assumptions, Ghost is still bounded from below by the accumulated noise energy, while LAI removes $Y$ by design and is empirically more stable.

# D    DETAILED INFORMATION ON DATASETS AND MODEL TRAINING

## D.1    DATASETS

We discuss datasets in Section 5 and 6 below.

### D.1.1    NLP DATASETS FOR LLM

For the four GLUE datasets—*SST2*, *MRPC*, *QNLI*, and *RTE* (Wang et al., 2018), we provide the following descriptions. The *Stanford Sentiment Treebank* (*SST2*) dataset consists of sentences labeled as positive or negative sentiment. It includes 67,349 training examples and 872 validation examples, making it a standard benchmark for sentiment classification tasks. The *Microsoft Research Paraphrase Corpus* (*MRPC*) dataset contains sentence pairs labeled as semantically equivalent or not. It includes 3,668 training examples and 408 validation examples. This dataset is widely used to evaluate paraphrase detection methods. The *Question Natural Language Inference* (*QNLI*) dataset is a large-scale corpus for question answering, derived from the Stanford Question Answering dataset. It consists of 104,743 training examples and 5,463 validation examples. The task involves determining whether the context sentence contains the answer to the question. The *Recognizing Textual Entailment* (*RTE*) dataset consists of sentence pairs labeled as entailment or not entailment. It includes 2,490 training examples and 277 validation examples. This dataset is derived from a series of annual textual entailment challenges and serves as a benchmark for textual entailment tasks.

### D.1.2    VISION DATASETS FOR CLASSIFICATIONS

Both the *CIFAR-10N* and *CIFAR-100N* datasets (Wei et al., 2022) consist of the same input images that make up the *CIFAR-10* (10 classes) and *CIFAR-100* (100 classes) datasets (Krizhevsky et al., 2009), respectively. Each input is a $32\times32$ RGB image with a dimension of $(3\times32\times32)$. However, for *CIFAR-10N* and *CIFAR-100N*, the labels are noisy, as they contain real-world human annotation errors collected using 3 annotators on Amazon Mechanical Turk. As these datasets are based on human-annotated noise, they model noisy real-world datasets more realistically, compared to synthetic data alternatives. The training set for both datasets contains 50,000 image-label pairs, and the test set contains 10,000 image-label pairs that are free from noise. For *CIFAR-10N* we utilize three

noise settings for experiments in the paper– (1) *Worst* is the dataset version with the highest noise rate (40.21%) as the worst possible annotation label for the image is chosen, (2) *Aggregate* is the least noisy dataset (9.03%) as labels are chosen via majority voting amongst the annotations, and (3) *Random* has intermediate noise (17.23%) and consists of picking one of the annotators' labels. We use the first annotator for the random labels. For *CIFAR-100N* there is only a single noisy setting due to the large number of labeling classes, and the overall noise rate is 40.20%.

### D.1.3 TEXT DATASETS FOR CLASSIFICATIONS

Both *20New-N* and *Emotion-N* datasets are derived from the original *20 Newsgroups* (20 classes) (Lang, 1995) and *Emotion* (6 classes) (Saravia et al., 2018) datasets, respectively. The inputs for these datasets consist of textual data, where each example corresponds to either a news article (*20 Newsgroups*) or an emotional text snippet (*Emotion*). However, for *20new-N* and *Emotion-N*, the labels are intentionally noisy, as 40% of the training set labels have been randomly replaced with incorrect labels. This noise was artificially introduced to simulate realistic label noise scenarios.

### D.2 MODELS AND METHODS

We now describe the models and the methods used in our experiments throughout the main paper. First, we describe the ResNet-18 (He et al., 2016) architecture used as the base model for the noisy vision datasets, then the BERT (Devlin, 2018) model for text datases. We also describe implementation details and parameter values and the influence-based baselines used throughout the paper.

### D.2.1 RESNET-18

The ResNet-18 model used in this study is implemented following the setup described in Section 4. The model is a convolutional neural network designed with 18 layers, based on the architecture proposed in He et al. (2016). It was trained on the *CIFAR-100N* dataset without pretraining on external datasets such as ImageNet. The training process adopts default parameters: a batch size of 512, an initial learning rate of $10^{-2}$, and the SGD optimizer with momentum (0.9) and weight decay ($5 \times 10^{-4}$). The model is trained over 150 epochs, with validation and test batch sizes set to 4000 and 1280, respectively. Experiments are conducted with different noise types specified for the dataset, including clean and noisy variants, as well as additional hyperparameters tailored to methods like SPL and IP, which are evaluated in this work.

### D.2.2 BERT

For both *20New-N* and *Emotion-N* dataset in Section 6, we use the BERT (Devlin, 2018) model for classification tasks. Key hyperparameters for training include a learning rate of $3 \times 10^{-5}$, a batch size of 32, and 3 training epochs. These parameters are consistent across experiments to ensure comparability of results under noisy label conditions.

### D.2.3 GPT-NEO

For the GLUE benchmark tasks, we fine-tuned the GPT-Neo-125M model (Gao et al., 2020) for sequence classification across tasks such as *SST-2*, *MRPC*, *QNLI*, and *RTE* with the number of classes varying based on the task. The training setup used a learning rate of $2 \times 10^{-5}$, a batch size of 16 for both training and evaluation, a weight decay of 0.01, and a maximum sequence length of 128 tokens. The number of epochs was set to 5 for all tasks, ensuring efficient fine-tuning.

### D.2.4 RETRAIN-BASED BASELINES

In our experiments, we utilize the following retrain-based methods as baselines: LiSSA (Koh and Liang, 2017) uses Hessian-vector products to approximate $H^{-1}$. DataInf (Kwon et al., 2023) applies an efficient closed-form surrogate for $H^{-1}$, TRAK (Park et al., 2023) employs randomly-projected kernel to reduce the dimensional of the Hessian matrix for tractable computing, and IP (Yang et al., 2024) replaces the Hessian matrix with the identity matrix. In addition, we consider three optimization-aware influence baselines, SGD-Influence (Hara et al., 2019), MoSo (Tan et al., 2023),

and Z0-Inf (Kokhlikyan et al., 2025). SGD-Influence estimates the effect of removing each training point by aggregating its contribution along the SGD trajectory and therefore requires access to (or storage of) intermediate checkpoints; MoSo approximates the leave-one-out risk by moving one sample out at a time through influence-style corrections on top of a reference trajectory; Z0-Inf further avoids explicit Hessian or Hessian-vector computations by using zeroth-order finite-difference perturbations of the parameters to approximate data influence. For a fair comparison, we implement these three methods under the same filter–then–retrain protocol as above, pruning samples with negative estimated influence and retraining the model on the remaining data.

### D.2.5 Curriculum-Based Baseline

For curriculum-style and online data valuation baselines, we include Self-Paced Learning (SPL) (Kumar et al., 2010), Ghost influence (Wang et al., 2024c), and our online variants LLI and LAI. SPL serves as a curriculum-learning baseline: after a short warm-up where all samples are used, it retains only those examples whose training loss is below a pace parameter. Ghost influence is the dynamic extension of Hessian-free influence that computes per-sample influence within each mini-batch by decomposing gradient inner products into layer-wise embedding similarities and feedback terms, then discarding samples with negative scores. LLI (Layer-aware Last-layer Influence) is a lightweight variant that scores samples using only last-layer embeddings and output gradients, while LAI (Layer-aware Influence) aggregates embedding similarities across all layers and reuses a shared output-layer gradient channel. Each method above operate in a single training run.

### D.3 Experimental Setup

#### D.3.1 Experimental Setup for LLMs

Fidelity validation and pre-training use the same setup: Adam ($\beta_1$=0.9, $\beta_2$=0.95, $\epsilon$=$10^{-8}$), learning rate $1\times10^{-4}$ with 500-step warmup and cosine decay, weight decay 0.01, gradient clip 1.0, context length 1,024.

#### D.3.2 Experimental Setup for Image and text datasets

To evaluate the effectiveness of our proposed method, we consider the classification with noisy labels. Specifically, we choose two widely used visual and two text datasets with label noise, *CIFAR-10N* (Wei et al., 2022), *CIFAR-100N* (Wei et al., 2022), *20News-N* (Lang, 1995), and *Emotion-N* (Saravia et al., 2018). *CIFAR-10N* encompasses three distinct noise settings: aggregate, random, and worst, denoted as "-*a*," "-*r*," and "-*w*," respectively. "*a*" means that labels are derived via majority voting among three annotators, with ties being resolved randomly, "*r*" adopts the label provided by the first annotator, while "*w*" selects the label from the least reliable annotator. For *20News-N* and *Emotion-N*, we introduce noise by randomly flipping 40% of training labels, aligning the noise level with that of *CIFAR-10N-w*. We use ResNet-18 (He et al., 2016) as the backbone model for visual datasets and train the model from scratch; for text datasets, we employ BERT (Devlin, 2018) as the backbone and add additional layer to fine-tune the whole network. In Sections 5 and 6, we randomly select 10% of validation set for each batch curation in vision datasets, and we randomly select 2,000 samples for each batch curation in text datasets.

## E Optimizer compatibility and practical guidance

LAI scores samples using raw last-layer gradients and multi-layer embedding similarities, which matches the one-step descent direction of SGD. For Adam and related adaptive methods, the update is a preconditioned gradient with first/second-moment statistics, so raw-gradient alignment may diverge slightly from the true update direction. A practical and low-overhead fix is to score in a diagonally preconditioned space: replace $\mathbf{g}^{(L)}$ by $\tilde{\mathbf{g}}^{(L)} = \mathbf{D}^{-1/2}\mathbf{g}^{(L)}$, where $\mathbf{D}$ uses layer- or block-level EMAs of squared gradients (an aggregated proxy of $\hat{v}_t$), and use $\tilde{\beta}^{(L)} = (\tilde{\mathbf{g}}_v^{(L)})^\top \tilde{\mathbf{g}}_i^{(L)}$ in Eq. (4). This reduces coordinate-scale mismatch without materializing per-sample moments. Achieving exact Adam-consistent scoring would require per-sample moment statistics and is typically too costly in complexity and memory.

