# OpenReview forum: "Layer-Aware Influence for Online Data Valuation Estimation"
_ICLR.cc/2026/Conference — Submitted to ICLR 2026_

### Official Review · Reviewer_j9uQ · 2025-10-25

**Soundness:** 2
**Presentation:** 3
**Contribution:** 2
**Rating:** 4
**Confidence:** 5

**Summary:**

This paper proposes Layer-Aware Influence (LAI), an online, Hessian-free method for estimating per-sample influence during training. LAI approximates influence by leveraging multi-layer embeddings but only requires gradients at the output layer, thereby avoiding costly full-network backpropagation. The method is designed to enable dynamic data curation (e.g., filtering out detrimental samples) within a single training pass, integrated naturally with SGD-style optimizers. The authors evaluate LAI across diverse settings and demonstrate improvements in accuracy, training efficiency, and influence fidelity compared to baselines like Ghost Influence and static influence estimators.

**Strengths:**

1. LAI significantly reduces computational and memory overhead compared to prior online influence methods (e.g., Ghost Influence), making dynamic data valuation feasible for large models like LLMs.

2. The experiments are extensive and cover multiple modalities (text, vision) and training regimes (pretraining, fine-tuning). The fidelity analysis against Monte Carlo Shapley values is particularly compelling.

3. The insight to replace per-layer gradient feedback with a single output-layer signal while preserving multi-layer embedding information is clever and empirically effective.

**Weaknesses:**

1. Overreliance on Implicit Convexity Assumptions: Despite claiming to operate in deep non-convex settings, the theoretical grounding of LAI (and its predecessor, Ghost Influence) still stems from second-order influence frameworks originally derived under local convexity or smoothness assumptions (e.g., Koh & Liang, 2017). The paper does not adequately address how these assumptions break down in highly non-convex, high-dimensional landscapes typical of deep learning. This limits the theoretical justification for why the inner-product-based influence proxy remains meaningful throughout training.

2. Moreover, during training, the key assumption in Koh et al.’s work, namely, the existence of a local stationary point. This is generally not satisfied. Consequently, their methodology is theoretically inapplicable within this framework.


3. While Appendix A provides a noise propagation analysis, it assumes structured perturbations (e.g., linearized backprop operators, bounded activations). In practice, modern architectures (with LayerNorm, residual connections, attention) violate these assumptions, and the claim that LAI “reduces variance” lacks rigorous non-asymptotic guarantees in realistic settings.

4. The experiment was too small in scale and lacked persuasiveness. Please provide the verification results under the Non-trivial experimental Settings of the large model training scenarios.

5. Missing **Key Related Works** on Non-Convex / Optimization-Aware Influence:

[a] Data Cleansing for Models Trained with SGD

[b] Data Pruning via Moving-one-Sample-out

[c].  Z0-Inf: Zeroth Order Approximation for Data Influence

**Questions:**

See Weakness.

I will raise my rate if my concerns are well addressed.

---

> ### Author Response · Authors · 2025-11-22
> **Response to Reviewer j9uQ (Part 1 of 3)**
>
> **We sincerely thank Reviewer j9uQ for the insightful and challenging questions.**
>
> **Assumptions of Influence Functions**
>
> We agree that the classical influence-function relies on assumptions such as local convexity, smoothness, and the existence of a well-defined stationary point. We also acknowledge that these assumptions are not strictly satisfied for modern deep neural networks. In fact, using influence-function–inspired quantities (or their first-order proxies) for deep models has already become standard practice in the community, although this is still an open question in the data valuation community. A large body of work applies influence-style or gradient-based influence estimates to deep networks in both vision and NLP, and evaluates them primarily through empirical effectiveness rather than strict satisfaction of the original convexity and stationarity assumptions [1-6].
>
> We would like to kindly point out that addressing these assumptions is not the research focus of our paper, i.e., our work does not aim to close this theoretical gap. Instead, our goal is to design a practical and scalable influence-based data valuation method for deep models, rather than to re-establish a fully rigorous influence-function theory under non-convex objectives. In this sense, our method follows this widely adopted line of work: we use the classical influence formulation as an intuition to motivate our design, but the core contribution of the paper lies in making influence-based data valuation efficient and effective for deep networks at scale. Even without strictly following these assumptions, we empirically demonstrate the promising performance of our LAI over several baselines. We will clarify this positioning and explicitly discuss the theoretical limitations of classical influence-function assumptions in the revision.
>
> **The experiment was too small**
> We appreciate the reviewer’s suggestion and would like to clarify that we indeed conducted experiments under non-trivial large-model scenarios: specifically, pre-training experiments with GPT-Neo 125M in Section 5.1 and fine-tuning experiments on GLUE benchmarks in Section 5.3.
>
> If Reviewer J9UQ still finds these experiments insufficient, we are eager to address their concerns regarding large-scale experiments by specifying the experiment types and datasets. We are happy to follow Reviewer J9UQ's suggestions and conduct additional experiments within our current computational resources, and provide our responses before the author-reviewer discussion period.

---

> > ### Author Response · Authors · 2025-11-22
> > **Response to Reviewer j9uQ (Part 2 of 3)**
> >
> > **Missing Key Related Works**. We thank the reviewer for pointing out these closely related optimization-aware influence methods, including SGD-Influence [7], MoSo [8], and Z0-Inf [9]. In the revision, we will explicitly discuss how these approaches relate to our online, layer-aware estimator in the *Related Work* section. In addition, we have implemented all three as baselines under our training setup, and their predictive performance and runtime/memory cost are reported in the comparison table below. Although these methods refine influence functions for static data valuation, a substantial performance gap remains when compared with online data valuation methods—namely, our LLI and LAI. This gap highlights the necessity of incorporating online data valuation during model optimization. Note that SGD-Influence requires substantial disk space to store all checkpoints. For 20News-N and Emotion-N, we estimate that SGD-Influence would require approximately 0.4 TB and 0.6 TB of disk space, respectively, which exceeds the available storage capacity on our workstation. Moreover, in the original SGD-Influence paper, the authors did not report experiments on mid-sized models.
> >
> > Table: Results of classification accuracy on benchmark datasets with noise labels. The methods with * are newly added during the author-reviewer discussion period. OOM means out of memory, and OOD means out of disk.
> >
> > | Methods        | *CIFAR-10N-a*      | *CIFAR-10N-r*      | *CIFAR-10N-w*      | *CIFAR-100N*       | *20News-N*         | *Emotion-N*        | Avg.   |
> > |----------------|--------------------|--------------------|--------------------|--------------------|--------------------|--------------------|--------|
> > | Vanilla        | 85.36 ± 0.18       | 83.19 ± 0.41       | 75.37 ± 1.05       | 40.87 ± 0.38       | 58.99 ± 0.92       | 84.79 ± 1.55       | 71.43  |
> > | LiSSA          | 78.67 ± 0.27       | 74.15 ± 0.44       | 60.44 ± 0.52       | 10.16 ± 0.41       | 63.41 ± 0.90       | **89.24 ± 1.16**   | 62.68  |
> > | DataInf        | 79.80 ± 0.41       | 75.34 ± 0.58       | 61.90 ± 1.53       | 30.69 ± 0.71       | 63.36 ± 0.37       | 89.08 ± 0.51       | 66.70  |
> > | TRAK           | 83.68 ± 0.25       | 80.94 ± 0.42       | 71.77 ± 0.62       | 32.93 ± 0.83       | OOM                | OOM                | 67.33  |
> > | IP             | 80.64 ± 0.48       | 78.84 ± 0.73       | 68.28 ± 0.98       | 30.82 ± 0.86       | **63.45 ± 0.75**   | 89.30 ± 0.54       | 68.56  |
> > | SGD-Influence* | 80.81 ± 0.00       | 81.75 ± 0.00       | 78.40 ± 0.00       | 30.36 ± 0.00       | OOD                | OOD                | 67.83  |
> > | MoSo*          | 71.74 ± 0.52       | 66.37 ± 0.70       | 38.73 ± 0.73       | 30.02 ± 0.58       | 61.68 ± 0.98       | 86.11 ± 0.95       | 59.11  |
> > | Z0-Inf*        | 81.30 ± 0.17       | 81.30 ± 0.30       | 74.82 ± 0.40       | 37.73 ± 0.61       | 63.41 ± 0.63       | 88.91 ± 0.57       | 72.35  |
> > | SPL            | 74.60 ± 2.06       | 74.42 ± 2.46       | 65.39 ± 1.41       | 41.96 ± 0.74       | 55.71 ± 1.33       | 79.66 ± 2.08       | 65.29  |
> > | Ghost          | **85.52 ± 0.31**   | 83.46 ± 0.59       | 75.98 ± 0.40       | 41.90 ± 0.63       | 63.22 ± 1.16       | 88.36 ± 0.44       | 73.07  |
> > | RHO-Loss*      | 11.83 ± 1.35       | 12.27 ± 1.63       | 11.44 ± 2.08       | 3.60 ± 0.81        | 60.24 ± 1.97       | 81.42 ± 2.28       | 30.13  |
> > | LLI (Ours)     | 84.13 ± 0.59       | 82.49 ± 0.16       | 75.87 ± 1.50       | **44.22 ± 0.56**   | 63.38 ± 0.86       | 88.64 ± 0.52       | 73.12  |
> > | LAI (Ours)     | 84.78 ± 0.52       | **83.69 ± 0.11**   | **76.43 ± 0.39**   | 44.06 ± 0.73       | 63.23 ± 0.97       | 88.45 ± 0.54       | **73.44** |
> >
> > **We kindly note that a revised version of our paper will be uploaded within the coming week.**

---

> > > ### Author Response · Authors · 2025-11-22
> > > **Response to Reviewer j9uQ (Part 3 of 3)**
> > >
> > > Reference
> > >
> > > [1] Pang Wei Koh and Percy Liang. Understanding black-box predictions via influence functions. In International Conference on Machine Learning, 2017.
> > >
> > > [2] Andrea Schioppa, Polina Zablotskaia, David Vilar, and Artem Sokolov. Scaling up influence functions. In AAAI Conference on Artificial Intelligence, 2022.
> > >
> > > [3] Ziao Yang, Han Yue, Jian Chen, and Hongfu Liu. Revisit, extend, and enhance hessian-free influence functions. arXiv preprint arXiv:2405.17490, 2024.
> > >
> > > [4] Roger Grosse, Juhan Bae, Cem Anil, Nelson Elhage, Alex Tamkin, Amirhossein Tajdini, Benoit Steiner, Dustin Li, Esin Durmus, Ethan Perez, et al. Studying large language model generalization with influence functions. arXiv preprint arXiv:2308.03296, 2023.
> > >
> > > [5] Yongchan Kwon, Eric Wu, Kevin Wu, and James Zou. Datainf: Efficiently estimating data influence in lora-tuned llms and diffusion models. arXiv preprint arXiv:2310.00902, 2023.
> > >
> > > [6] Zayd Hammoudeh and Daniel Lowd. Training data influence analysis and estimation: A survey. arXiv preprint arXiv:2212.04612, 2022.
> > >
> > > [7] Satoshi Hara, Atsushi Nitanda, and Takanori Maehara. Data cleansing for models trained with SGD. In Advances in Neural Information Processing Systems, 2019.
> > >
> > > [8] Haoru Tan, Sitong Wu, Fei Du, Yukang Chen, Zhibin Wang, Fan Wang, and Xiaojuan Qi. Data pruning via moving-one-sample-out. In Advances in Neural Information Processing Systems, 2023.
> > >
> > > [9] Narine Kokhlikyan, Kamalika Chaudhuri, and Saeed Mahloujifar. Z0-Inf: Zeroth order approximation for data influence. arXiv preprint arXiv:2510.11832, 2025.

---

### Official Review · Reviewer_Jo6W · 2025-10-27

**Soundness:** 4
**Presentation:** 3
**Contribution:** 3
**Rating:** 6
**Confidence:** 3

**Summary:**

This submission focuses on data-centric learning methods where the focus is on careful selection of training data samples instead of improving the architecture. The major challenge in such methods is the efficiency with which such methods can be designed.
In this work, authors propose a layer-aware online estimator that avoids need for parameter-lever or full-network gradients.
The method mainly focuses on online dynamic per-sample influence estimation, which is noticeably suffers from inverse Hessian computation. To that end, this work introduces layer-aware influence (LAI) estimator that uses a single, stable feedback channel.

**Strengths:**

Strengths:
- Unlike previous methods, this approach focuses on online data valuation
- Computes online per-sample inflated during training and integrates it naturally with SGD-style updates
- Identifies the issues where use on inverse of Hessian matrix and oversight due to no-longer-be-deterimental samples for the trained model
- Focuses on beneficial-or-not approach instead of easy-to-hard (curriculum) learning approach
- Experiments are very carefully done for image and text classification

**Weaknesses:**

Weaknesses:
- It is not clear how existing approaches are different from importance-driven methods like Salaun et al. 2025a,b for data pruning or careful data selection
- The overall theoretical contribution of the work seems quite limited. However, the simple adjustments done writ Wang et al. 2024c  and making it layer-aware seems to bring noticeable improvements in both image and text classifications. This needs to be carefully explained.

**Questions:**

Questions:
- Are influence functions different from importance functions (weights) used by existing literature?
- How to handle cases where multiple influence functions are applicable? Does it resembles Multiple importance Sampling methods from the probability statistics (Salaun et al. 2025b)?
- Salaun et al. 2025a work on online importance sampling using weights from the last layer to design importance weighting function. Are there any benefits such methods bring to sample influence estimation?

Online Importance Sampling for Stochastic Gradient Optimization, Salaun et al. 2025a
Multiple Importance Sampling for Stochastic Gradient Estimation, Salaun et al. 2025b

---

> ### Author Response · Authors · 2025-11-22
> **Response to Reviewer Jo6W (Part 1 of 3)**
>
> **We sincerely thank Reviewer Jo6W for the thoughtful and detailed feedback, especially on the connections to importance-driven methods.**
>
> **Importance-Driven Pruning vs. Our Approach**
> We thank the reviewer for pointing out the connection to importance-driven methods such as Salaun's papers [1-2].
>
> - **Conceptually**, importance-driven methods and our approach address different research problems. GradientIS and GradientMIS [1-2] are gradient-based importance sampling schemes for SGD whose primary goal is to **reduce the variance of the stochastic gradient and thereby speed up optimization**. They adapt sampling probabilities according to the norm of the output-layer gradient, and optionally prune samples with persistently small gradient norms, that is, samples that contribute little to the training gradient. In particular, these methods do not estimate per-example influence on a held-out objective and do not use any validation signal. Our work instead targets online data valuation, i.e., by **curating suitable sample to improve the model performance**. We estimate the data valuation of the samples in each batch, and just involve the beneficial samples for model optimization.
>
> - **Methodologically**, our estimator is based on inner products between training and validation gradients and a layer-wise influence proxy, while [1-2] design gradient-norm-based importance weights to reduce the variance of SGD updates and do not approximate influence.
>
> We agree that these works are relevant and that the difference to our setting should be stated more clearly. We will clarify this relationship in the revised version by adding a dedicated paragraph in the related work section that discusses GradientIS/GradientMIS and contrasts them with our online data valuation framework.
>
> **Limited Theoretical Contribution**. Regarding the concern that the overall theoretical contribution seems limited, we would like to clarify that we conducted theoretical analyses in multiple parts of the paper. These analyses appear in the following sections, which together form a coherent theoretical story. We first analyze how ghost influence behaves along the backpropagation chain, then model LAI as a structured approximation to ghost influence, and finally compare their bias–variance tradeoffs to explain why the simplified layer-aware estimator yields more stable and accurate data valuation.
>
> - **Understanding the Backpropagation Chain Underlying Ghost Influence**. Section 3.2 together with Appendix A analyzes the backpropagation chain used by [3]. We decompose the inner product between two sample gradients into per-layer embedding similarities and per-layer feedback terms, and show that ghost influence accumulates stochastic perturbations at every layer. This explains why ghost influence can have high variance and unstable rankings in deep networks.
>
> - **Modeling LAI as a Structured Approximation to Ghost Influence**. Section 4.1 and Appendix B formally present our Layer-Aware Influence (LAI) estimator as a structured approximation to ghost influence. By replacing the full set of per-layer feedback channels $\{\beta^{(l)}\}_{l=1}^{L}$ with a single output-layer channel $\beta^{(L)}$, we derive an explicit upper bound on $\lvert I_{\text{Ghost}}(z_{j}) - I_{\text{LAI}}(z_{j}) \rvert$ under mild assumptions on gradient decay and bounded activations. This shows that the layer-aware modification is a controlled and theoretically grounded approximation rather than an ad hoc tweak.
>
> - **Comparison between Ghost Influence and LAI in terms of Bias–Variance Tradeoffs**. Appendix C provides a bias–variance comparison between ghost influence and LAI. The analysis shows that LAI trades a small and controlled bias for a significant reduction in variance by removing the depth-wise accumulation of noise present in ghost influence. This theoretically explains why the simple layer-aware adjustment yields improved data valuation quality.
>
> In addition, Section 4.2 discusses the computational and memory advantages of LAI over ghost influence and explains why using a single, more stable feedback channel leads to more reliable online curation. Sections 5 and 6 empirically confirm that these theoretically motivated changes yield consistent gains on both image and text classification tasks compared to [3], under matched training and curation protocols.
>
> In the revised version, we will add a short paragraph at the end of Section 4 that explicitly summarizes (i) the key conceptual differences between LAI and [3], and (ii) how our theoretical analyses explain the observed improvements on both vision and NLP benchmarks.

---

> > ### Author Response · Authors · 2025-11-22
> > **Response to Reviewer Jo6W (Part 2 of 3)**
> >
> > **Difference Between Influence Functions vs. Importance Weights**
> > We thank the reviewer for raising this question. We agree that the terminology can be confusing, but influence functions and importance weights play different roles. In the following, we briefly clarify the distinction between influence functions and importance weights from their definitions and purposes:
> >
> > - **Influence functions** quantify how an infinitesimal upweighting or removal of a single training example would change a downstream objective such as validation loss. They are per-example effect measures on a held-out metric, typically derived from the gradient of the validation loss and (an approximation of) the inverse Hessian. The main goal is to value training data and understand how individual points affect generalization, robustness, or fairness.
> >
> > - **Importance weights**, in contrast, are the weights used in importance sampling or non-uniform mini-batch construction within SGD. They specify how often and how strongly each training point enters the stochastic gradient estimate, and they are designed to reduce the variance of SGD updates or to speed up optimization. Works such as [1-2] construct gradient-norm-based importance weights purely from training loss and training gradients; they do not approximate influence on a held-out objective.
> >
> > Our method is built on the influence-function viewpoint: we estimate the effect of each training example on a validation objective via a layer-aware, Hessian-free approximation based on inner products between training and validation gradients. These influence scores can then be used for online data valuation and curation, and could in principle also serve as importance weights inside importance-sampling schemes. This makes our approach complementary rather than equivalent to existing importance-driven methods. Exploring this connection more systematically, for example by directly using our influence scores as importance weights in adaptive sampling, is an interesting direction for future work, and we will briefly highlight this possibility in the revised version.
> >
> > **Multiple Goals of Influence Functions vs. Importance Sampling**
> > We thank the reviewer for this question. In short, in the area of influence functions [1], multiple influence functionals typically arise from multiple downstream objectives (e.g., the sample influence on model performance in terms of utility, robustness, or fairness), and they are combined using standard multi-objective weighting. This is conceptually different from multiple importance sampling (MIS) as used by [2], which combines several proposal distributions and importance weights for Monte Carlo estimation.
> >
> > More concretely, in influence function area, having multiple influence functions means that there are several validation objectives or evaluation metrics of interest (for example, a standard accuracy loss, a robustness loss, and a fairness penalty). Each such objective induces its own influence functional, that is, a mapping that assigns a per-example influence score with respect to that particular objective. When several objectives matter at the same time, we follow common practice in multi-objective learning and form a single combined validation objective as a weighted combination of the individual objectives, where the weights encode application-specific trade-offs between metrics. Since the influence function is linear in the objective, the influence of a training example on the combined objective is simply the same weighted combination of its influences on each individual objective. This provides a principled and unambiguous way to handle multiple influence functions. In our experiments we instantiate this framework with a single primary validation loss, but the same formulation directly extends to multiple metrics.
> >
> > Therefore, our setting is different in nature from the multiple importance sampling methods studied by [2]. MIS combines several proposal distributions and associated importance weights in order to obtain an unbiased, low-variance estimator of an integral and to control the variance of stochastic gradient updates. In contrast, our multiple influence functions are not proposal distributions but deterministic per-example effect measures for different downstream objectives, and combining them corresponds to scalarizing a multi-objective validation criterion rather than mixing sampling distributions.

---

> > > ### Author Response · Authors · 2025-11-22
> > > **Response to Reviewer Jo6W (Part 3 of 3)**
> > >
> > > **Benefits of online importance sampling for influence-based estimation**
> > > We thank the reviewer for this question. In short, we view online importance sampling methods such as [1] as highly complementary to influence-based approaches, and we believe their gradient-based weighting schemes are very relevant for future influence work.
> > >
> > > In our paper, the research question is centered on online data valuation: we develop a layer-aware, Hessian-free influence approximation and use it primarily for digital sample curation within each mini-batch, that is, ranking and selecting examples based on their estimated influence. We did not change the per-sample loss weights inside the batch.
> > >
> > > However, we agree that there is a natural connection: influence scores could also be turned into per-sample weights inside each batch and combined with importance-weighting ideas similar to [1-2] to define influence-aware importance functions. This could potentially bring the variance-reduction benefits of importance sampling into influence-based data valuation. Exploring such influence-weighted sampling schemes is an interesting direction for future work, and we will mention this explicitly in the revised version.
> > >
> > > Exploring whether ideas from MIS could nevertheless help design better ways of combining several influence signals, for example from different objectives or layers, is an interesting direction for future work, and we will briefly mention this possible connection in the revised version.
> > >
> > > **We kindly note that a revised version of our paper will be uploaded within the coming week.**
> > >
> > > Reference
> > >
> > > [1] Corentin Salaün, Xingchang Huang, Iliyan Georgiev, Niloy J. Mitra, and Gurprit Singh. Online importance sampling for stochastic gradient optimization. In International Conference on Pattern Recognition Applications and Methods, 2025.
> > >
> > > [2] Corentin Salaün, Xingchang Huang, Iliyan Georgiev, Niloy J. Mitra, and Gurprit Singh. Multiple importance sampling for stochastic gradient estimation. In International Conference on Pattern Recognition Applications and Methods, 2025.
> > >
> > > [3] Jiachen T. Wang, Prateek Mittal, Dawn Song, and Ruoxi Jia. Data Shapley in one training run. arXiv preprint arXiv:2406.11011, 2024.

---

### Official Review · Reviewer_6gd8 · 2025-11-01

**Soundness:** 2
**Presentation:** 3
**Contribution:** 2
**Rating:** 4
**Confidence:** 4

**Summary:**

This paper proposes Layer-Aware Influence (LAI), a Hessian-free, online data valuation method that dynamically estimates per-sample influence during training without requiring full backpropagation or retraining. LAI leverages multi-layer embeddings but only computes gradients up to the output layer, significantly reducing computational and memory overhead compared to prior influence estimation methods like Ghost Influence. The approach is evaluated across diverse settings, including LLM pre-training, LLM fine-tuning, and image/text classification with noisy labels, and demonstrates consistent improvements in accuracy, validation loss, and training efficiency while maintaining high fidelity to Monte Carlo Shapley values.

**Strengths:**

Novelty and Practicality: LAI introduces a lightweight yet effective approximation to online influence estimation by avoiding per-layer per-sample gradients. This design choice addresses a critical scalability bottleneck in dynamic data curation, making it feasible for large models like LLMs.

Strong Empirical Validation: The method is rigorously evaluated across multiple modalities (text and vision), training regimes (pre-training and fine-tuning), and noise settings. Results consistently show that LAI outperforms both static influence methods and recent online baselines (e.g., Ghost Influence) in both performance and efficiency.

Theoretical Justification: The paper provides a clear bias-variance analysis showing that LAI reduces noise accumulation compared to Ghost Influence, explaining its superior empirical stability and fidelity. The derivation in the appendix further supports LAI as a principled approximation.

**Weaknesses:**

Validation Set Dependency: The method relies on a validation set for influence estimation, which may not always be available (e.g., in unsupervised pre-training). While the paper uses self-influence as a workaround, this approach’s robustness and generalizability remain underexplored.


---

Lack of Comparison to Non-Convex, Non-Influence-Based Data Curation Methods: The paper does not compare against recent dynamic sample selection strategies that do not rely on influence functions or convex assumptions (e.g., JEST, ACID, or RHO-Loss variants), which are relevant baselines given the paper’s focus on online curation.

---

Missing Key Related Works on Non-Convex / Optimization-Aware Influence:

[a] Data Cleansing for Models Trained with SGD

[b] Data Pruning via Moving-one-Sample-out

[c]. Z0-Inf: Zeroth Order Approximation for Data Influence

---

This method inherits the formulation from classical Koh's influence functions, which rely on assumptions such as local convexity, smoothness, and the existence of a well-defined stationary point. This represents a major theoretical flaw in the paper, because such a condition rarely holds in deep learning.

The paper does not rigorously analyze how or whether these assumptions hold during training, nor does it provide alternative theoretical grounding for why the inner-product-based influence proxy remains valid in such settings.

**Questions:**

See weakness

---

> ### Author Response · Authors · 2025-11-21
> **Response to Reviewer 6gd8 (Part 1 of 3)**
>
> **We sincerely thank Reviewer 6gd8 for the careful reading of our paper and the constructive comments and suggestions.**
>
> **Validation Set Dependency.** We agree with Reviewer 6GD8 that the influence function-based methods rely on the validation set for data validation, which is a standard routine practice in the community [1-6]. The self-influence variant is designed for regimes where a labeled validation set is unavailable (e.g., large-scale pre-training). Recent work shows that self-influence is a stable and useful signal for data curation, and that self-influence-based filtering improves downstream performance on machine translation, question answering and text classification benchmarks [7].
>
> In our paper, we provide both settings with and without the validation set. In our LLM pre-training experiments, where validation set is not provided, we measure the expected squared gradient norm of each training point $z$ across training checkpoints,
>
> $$
> \sum_i \eta_i \|\nabla_w \ell(z; w_i)\|_2^2
> $$
>
> Examples with consistently large self-influence induce large and unstable updates and are therefore more likely to be noisy or outlying, which hurts generalization; we down-weight or remove such points in the same spirit as the validation-based influence variant. We will add these references and explicitly clarify that our self-influence variant is a validation-free, robust data valuation mechanism grounded in the same influence-based principle.
>
> **Non-Influence-Based Data Curation Methods.**
> We appreciate the reviewer for pointing out the line of work on dynamic sample selection beyond influence functions, including JEST, ACID and RHO-Loss. In our revision, we have added RHO-Loss [8] as a representative non-influence-based online curation baseline, since to the best of our knowledge it is the only method among these with publicly available code that can be directly adapted to our setting. As reported in the table below, the performance of RHO-Loss in our online noisy-label classification setup is noticeably lower than most other methods. We have carefully reproduced its implementation and hyperparameters following the original paper and official repository, so these numbers reflect its behavior in our specific per-batch curation with real or synthetic label noise, rather than implementation issues. We speculate that this performance gap mainly arises because RHO-Loss is not particularly well-suited to noisy-label learning in our online curation setting. In the revised version, we will also expand the Related Work section to more thoroughly discuss JEST, ACID/ACED, and the broader family of non-influence-based data curation methods, and clarify how our research question on online influence estimation is complementary to these lines of work.

---

> > ### Author Response · Authors · 2025-11-22
> > **Response to Reviewer 6gd8 (Part 2 of 3)**
> >
> > **Missing key related works on non-convex / optimization-aware influence.**
> > We thank the reviewer for pointing out these closely related optimization-aware influence methods, including SGD-Influence [9], MoSo [10], and Z0-Inf [11]. In the revision, we will explicitly discuss how these approaches relate to our online, layer-aware estimator in the *Related Work* section. In addition, we have implemented all three as baselines under our training setup, and their predictive performance and runtime/memory cost are reported in the comparison table below. Although these methods refine influence functions for static data valuation, a substantial performance gap remains when compared with online data valuation methods—namely, our LLI and LAI. This gap highlights the necessity of incorporating online data valuation during model optimization. Note that SGD-Influence requires substantial disk space to store all checkpoints. For 20News-N and Emotion-N, we estimate that SGD-Influence would require approximately 0.4 TB and 0.6 TB of disk space, respectively, which exceeds the available storage capacity on our workstation. Moreover, the original SGD-Influence paper focuses on relatively small-scale models and does not provide results for mid-sized architectures comparable to those used in our experiments.
> >
> > **Table.** Results of classification accuracy on benchmark datasets with noisy labels. The methods with `*` are newly added during the author–reviewer discussion period. OOM means out of memory, and OOD means out of disk.
> >
> > | Methods           | *CIFAR-10N-a*      | *CIFAR-10N-r*      | *CIFAR-10N-w*      | *CIFAR-100N*       | *20News-N*         | *Emotion-N*        | Avg.   |
> > |-------------------|--------------------|--------------------|--------------------|--------------------|--------------------|--------------------|--------|
> > | Vanilla           | 85.36 ± 0.18       | 83.19 ± 0.41       | 75.37 ± 1.05       | 40.87 ± 0.38       | 58.99 ± 0.92       | 84.79 ± 1.55       | 71.43  |
> > | LiSSA             | 78.67 ± 0.27       | 74.15 ± 0.44       | 60.44 ± 0.52       | 10.16 ± 0.41       | 63.41 ± 0.90       | **89.24 ± 1.16**   | 62.68  |
> > | DataInf           | 79.80 ± 0.41       | 75.34 ± 0.58       | 61.90 ± 1.53       | 30.69 ± 0.71       | 63.36 ± 0.37       | 89.08 ± 0.51       | 66.70  |
> > | TRAK              | 83.68 ± 0.25       | 80.94 ± 0.42       | 71.77 ± 0.62       | 32.93 ± 0.83       | OOM                | OOM                | 67.33  |
> > | IP                | 80.64 ± 0.48       | 78.84 ± 0.73       | 68.28 ± 0.98       | 30.82 ± 0.86       | **63.45 ± 0.75**   | 89.30 ± 0.54       | 68.56  |
> > | **SGD-Influence\*** | 80.81 ± 0.00     | 81.75 ± 0.00       | 78.40 ± 0.00       | 30.36 ± 0.00       | OOD                | OOD                | 67.83  |
> > | **MoSo\***        | 71.74 ± 0.52       | 66.37 ± 0.70       | 38.73 ± 0.73       | 30.02 ± 0.58       | 61.68 ± 0.98       | 86.11 ± 0.95       | 59.11  |
> > | **Z0-Inf\***      | 81.30 ± 0.17       | 81.30 ± 0.30       | 74.82 ± 0.40       | 37.73 ± 0.61       | 63.41 ± 0.63       | 88.91 ± 0.57       | 72.35  |
> > | SPL               | 74.60 ± 2.06       | 74.42 ± 2.46       | 65.39 ± 1.41       | 41.96 ± 0.74       | 55.71 ± 1.33       | 79.66 ± 2.08       | 65.29  |
> > | Ghost             | **85.52 ± 0.31**   | 83.46 ± 0.59       | 75.98 ± 0.40       | 41.90 ± 0.63       | 63.22 ± 1.16       | 88.36 ± 0.44       | 73.07  |
> > | **RHO-Loss\***    | 11.83 ± 1.35       | 12.27 ± 1.63       | 11.44 ± 2.08       | 3.60 ± 0.81        | 60.24 ± 1.97       | 81.42 ± 2.28       | 30.13  |
> > | LLI (Ours)        | 84.13 ± 0.59       | 82.49 ± 0.16       | 75.87 ± 1.50       | **44.22 ± 0.56**   | 63.38 ± 0.86       | 88.64 ± 0.52       | 73.12  |
> > | **LAI (Ours)**    | 84.78 ± 0.52       | **83.69 ± 0.11**   | **76.43 ± 0.39**   | 44.06 ± 0.73       | 63.23 ± 0.97       | 88.45 ± 0.54       | **73.44** |

---

> > > ### Author Response · Authors · 2025-11-22
> > > **Response to Reviewer 6gd8 (Part 3 of 3)**
> > >
> > > **Assumptions of Influence Functions.**
> > > We agree that the classical influence-function relies on assumptions such as local convexity, smoothness, and the existence of a well-defined stationary point. We also acknowledge that these assumptions are not strictly satisfied for modern deep neural networks. In fact, using influence-function–inspired quantities (or their first-order proxies) for deep models has already become standard practice in the community, although this is still an open question in the data valuation community. A large body of work applies influence-style or gradient-based influence estimates to deep networks in both vision and NLP, and evaluates them primarily through empirical effectiveness rather than strict satisfaction of the original convexity and stationarity assumptions [2-5].
> > >
> > > We would like to kindly point out that addressing these assumptions is not the research focus of our paper, i.e., our work does not aim to close this theoretical gap. Instead, our goal is to design a practical and scalable influence-based data valuation method for deep models, rather than to re-establish a fully rigorous influence-function theory under non-convex objectives. In this sense, our method follows this widely adopted line of work: we use the classical influence formulation as an intuition to motivate our design, but the core contribution of the paper lies in making influence-based data valuation efficient and effective for deep networks at scale. Even without strictly following these assumptions, we empirically demonstrate the promising performance of our LAI over several baselines. We will clarify this positioning and explicitly discuss the theoretical limitations of classical influence-function assumptions in the revision.
> > >
> > > **We kindly note that a revised version of our paper will be uploaded within the coming week.**
> > >
> > > **Reference**
> > >
> > > [1] Pang Wei Koh and Percy Liang. Understanding black-box predictions via influence functions. In International Conference on Machine Learning, 2017.
> > >
> > > [2] Andrea Schioppa, Polina Zablotskaia, David Vilar, and Artem Sokolov. Scaling up influence functions. In AAAI Conference on Artificial Intelligence, 2022.
> > >
> > > [3] Ziao Yang, Han Yue, Jian Chen, and Hongfu Liu. Revisit, extend, and enhance hessian-free influence functions. arXiv preprint arXiv:2405.17490, 2024.
> > >
> > > [4] Roger Grosse, Juhan Bae, Cem Anil, Nelson Elhage, Alex Tamkin, Amirhossein Tajdini, Benoit Steiner, Dustin Li, Esin Durmus, Ethan Perez, et al. Studying large language model generalization with influence functions. arXiv preprint arXiv:2308.03296, 2023.
> > >
> > > [5] Yongchan Kwon, Eric Wu, Kevin Wu, and James Zou. Datainf: Efficiently estimating data influence in lora-tuned llms and diffusion models. arXiv preprint arXiv:2310.00902, 2023.
> > >
> > > [6] Zayd Hammoudeh and Daniel Lowd. Training data influence analysis and estimation: A survey. arXiv preprint arXiv:2212.04612, 2022.
> > >
> > > [7] Irina Bejan, Artem Sokolov, and Katja Filippova. Make every example count: On the stability and utility of self-influence for learning from noisy NLP datasets. In Conference on Empirical Methods in Natural Language Processing, 2023.
> > >
> > > [8] Sören Mindermann, Jan M Brauner, Muhammed T Razzak, Mrinank Sharma, Andreas Kirsch, Winnie Xu, Benedikt Höltgen, Aidan N Gomez, Adrien Morisot, Sebastian Farquhar, et al. Prioritized training on points that are learnable, worth learning, and not yet learnt. In International Conference on Machine Learning, 2022.
> > >
> > > [9] Satoshi Hara, Atsushi Nitanda, and Takanori Maehara. Data cleansing for models trained with SGD. In Advances in Neural Information Processing Systems, 2019.
> > >
> > > [10] Haoru Tan, Sitong Wu, Fei Du, Yukang Chen, Zhibin Wang, Fan Wang, and Xiaojuan Qi. Data pruning via moving-one-sample-out. In Advances in Neural Information Processing Systems, 2023.
> > >
> > > [11] Narine Kokhlikyan, Kamalika Chaudhuri, and Saeed Mahloujifar. Z0-Inf: Zeroth order approximation for data influence. arXiv preprint arXiv:2510.11832, 2025.

---

### Author Response · Authors · 2025-11-26
**Summary of Revisions in the Revised Manuscript**

Dear Reviewers,

Thank you again for your valuable time and thoughtful feedback, which have greatly improved the quality of our paper. Below we summarize the major revisions made in response to your comments:

1. **Expanded Related Work.**
   We substantially enriched the related work section (Lines 108–126 and 146–155) to provide clearer context and more comprehensive positioning of our contributions.

2. **Additional Baselines.**
   We added three new baseline methods [1–3] to Table 3 for more thorough comparisons.

3. **Appendix Summary.**
   We included a new summary of the appendix to clarify how the three components of our theoretical analysis connect and collectively support our main results.

4. **Limitation and Future Work.**
   We added a new section titled “Limitation and Future Direction” that discusses the theoretical assumptions, approximation limits, and future extensions of our LAI framework.

5. **Acknowledgments.**
   We added an acknowledgments section to explicitly thank the reviewers for their constructive feedback and helpful suggestions, which have substantially improved the clarity and completeness of our work.

**References**

[1] Data Cleansing for Models Trained with SGD

[2] Data Pruning via Moving-one-Sample-out

[3] Z0-Inf: Zeroth Order Approximation for Data Influence


Kind regards,
Authors

---

### Meta-Review · Area_Chair_o1Wr · 2026-01-05

**Summary:**

1. The method is motivated by influence functions, which rely on assumptions (local convexity, smoothness, stationary points) that are generally violated in deep learning. Reviewers questioned whether the proposed inner-product–based proxy remains principled under such settings.

2. Reviewers viewed LAI largely as a layer-aware simplification of prior ghost influence methods rather than a fundamentally new influence formulation, with insufficient theoretical justification.

3. The distinction between influence-based valuation and recent importance-driven online data selection methods was not initially clear, raising concerns about conceptual overlap.

4. Lack of comparisons and discussions with recent works.

**Reviewer Concerns:**

1. The authors added comparisons with SGD-Influence, MoSo, Z0-Inf, and RHO-Loss, and expanded the related work accordingly. This substantially improved empirical completeness.

2. The rebuttal clearly explained runtime and memory advantages over ghost influence and optimization-aware baselines, supported by extensive experiments across LLMs and vision benchmarks.

3. Additional experiments and analysis strengthened the claim that LAI is empirically stable and effective, especially in noisy-label settings and large-model regimes.

Concerns still outstanding:

1. The rebuttal explicitly acknowledges that classical influence-function assumptions do not hold and states that addressing this gap is not the goal of the paper. While this honesty is appreciated, it leaves the core concern unresolved: why the proposed influence proxy should be trusted beyond empirical correlation. No alternative theoretical framework is provided.

2. Despite strong engineering contributions, the method remains close to prior ghost influence and last-layer importance approaches. The rebuttal argues practicality rather than novelty, which may be insufficient for acceptance at ICLR.

3. Although the authors added discussion, the rebuttal does not convincingly articulate a clear conceptual separation between influence estimation and online importance weighting, especially when both reduce to gradient-based scoring.

**Reviewer Scores:**

1. Reviewer 6gd8 (initial rating 4)
Likely unchanged as core theoretical concerns remain unresolved despite added baselines.

2. Reviewer Jo6W (intial rating 6)
Likely unchanged or lower the rating. Improved clarity and experiments help, but limited theoretical novelty tempers enthusiasm.

3. Reviewer  j9uQ  (Initial rating 4)
Likely marginal shifts, but not enough to change overall outcome.

---

### Decision · Program_Chairs · 2026-01-26

Reject